Manuscript prepared for Atmos. Chem. Phys.
with version 2015/04/24 7.83 Copernicus papers of the LaTeX class copernicus.cls.
Date: 14 February 2017

# Wave Modulation of the Extratropical Tropopause Inversion Layer

Robin Pilch Kedzierski[1], Katja Matthes[1,2], and Karl Bumke[1]

[1]Marine Meteorology Department, GEOMAR Helmholtz Centre for Ocean Research Kiel, Kiel, Germany.
[2]Faculty of Mathematics and Natural Sciences, Christian-Albrechts-Universität zu Kiel, Kiel, Germany.

*Correspondence to:* Robin Pilch Kedzierski (rpilch@geomar.de)

**Abstract.**

This study aims to quantify how much of the observed strength and variability of the zonal-mean extratropical Tropopause Inversion Layer (TIL) comes from the modulation of the temperature field and its gradients around the tropopause by planetary and synoptic-scale waves. By analyzing high-resolution observations, it also puts other TIL enhancing mechanisms into context.

Using gridded Global Positioning System radio occultation (GPS-RO) temperature profiles from the COSMIC mission (2007-2013), we are able to extract the extratropical wave signal by a simplified wavenumber-frequency domain filtering method, and to quantify the resulting TIL enhancement. By subtracting the extratropical wave signal, we show how much of the TIL is associated with other processes, at mid and high latitudes, for both Hemispheres and all seasons.

The transient and reversible modulation by planetary and synoptic-scale waves is almost entirely responsible for the TIL in mid-latitudes. This means that wave-mean flow interactions, inertia-gravity waves or the residual circulation are of minor importance for the strength and variability of the mid-latitude TIL.

At polar regions, the extratropical wave modulation is dominant for the TIL strength as well, but there is also a clear fingerprint from sudden stratospheric warmings (SSWs) and final warmings in both hemispheres. Therefore, polar vortex breakups are partially responsible for the observed polar TIL strength in winter (if SSWs occur) and spring. Also, part of the polar summer TIL strength cannot be explained by extratropical wave modulation.

We suggest that our wave modulation mechanism integrates several TIL enhancing mechanisms proposed in previous literature while robustly disclosing the overall outcome of the different processes involved. Our study identifies which mechanisms dominate the extratropical TIL strength and their relative contribution, by analyzing observations only. It remains to be determined, however, which roles the different planetary and synoptic-scale wave types play within the total extratropical wave modulation of the TIL; and what causes the observed amplification of extratropical waves near the tropopause.

# 1   Introduction

The extratropical Tropopause Inversion Layer (TIL) is a strong temperature inversion at the extratropical tropopause with a corresponding static stability maximum right above. It is a fine-scale feature discovered via tropopause-based averaging (Birner et al., 2002; Birner, 2006), consisting of a thin layer of about 1km depth. Satellite Global Positioning System radio occultation observations (GPS-RO) show that the TIL is present globally (Grise et al., 2010).

The TIL is established as an important feature of the extratropical upper troposphere and lower stratosphere (UTLS) (Gettelman et al., 2011), and it is of interest to the scientific community for the following reasons: high static stability values theoretically affect the dispersion relations of atmospheric waves like Rossby or Inertia-Gravity waves since this parameter is part of different wave theory approximations (see Birner (2006); Grise et al. (2010) and references therein). In an idealized model experiment, Sjoberg and Birner (2014) showed that the TIL acts as a partial barrier for upward wave propagation. The study by Zhang et al. (2015) supports this hypothesis by showing inhibited upward propagation of Inertia-Gravity waves (IGW) due to the TIL, with data from a single US radiosonde station. Also, the TIL is likely to inhibit the cross-tropopause exchange of chemical compounds: high static stability suppresses vertical motion and is correlated with strong trace gas gradients (Hegglin et al., 2009; Kunz et al., 2009; Schmidt et al., 2010). In the next paragraphs we shortly review what is known so far about the observed variability of the extratropical TIL and the mechanisms proposed for its formation/enhancement.

Climatological studies about the seasonal, zonal-mean state of the TIL show that it reaches maximum strength during polar summer while it displays a weaker relative maximum in winter mid-latitudes (Birner, 2006; Randel et al., 2007; Randel and Wu, 2010; Grise et al., 2010). From a synoptic-scale perspective, the TIL in mid-latitude winter has very pronounced zonal structures, and the TIL within ridges (anticyclones) in mid-latitude winter has the same strength or even higher than any TIL observed in polar summer (Pilch Kedzierski et al., 2015). The cyclone-anticyclone modulation with weaker-stronger TIL is found at all extratropical latitudes and seasons (Randel et al., 2007; Randel and Wu, 2010; Pilch Kedzierski et al., 2015).

Different mechanisms are responsible for the formation/maintenance of the extratropical TIL:

- Radiative cooling below the tropopause by water vapor due to its strong gradient across the tropopause acts to enhance the TIL (Randel et al., 2007; Hegglin et al., 2009; Kunz et al., 2009; Randel and Wu, 2010), and a high-resolution model study by Miyazaki et al. (2010b, a) showed that radiative effects are dominant in polar summer, while dynamics enhance the TIL otherwise. Other modelling studies suggest that radiation and diabatic processes related to water vapor and clouds could share importance with dynamics in strengthening the mid-latitude TIL (Ferreira et al., 2016; Kunkel et al., 2016).

- The downwelling branch of the stratospheric residual circulation was proposed to cause dynamical heating above the tropopause and TIL enhancement in a model experiment by Birner (2010).

The first evidence of this was found by Wargan and Coy (2016) at high latitudes following major

sudden stratospheric warmings (SSWs), mainly caused by the convergence of the vertical component of the residual circulation ($\overline{w*}$). During a major SSW there is an acceleration of the residual circulation, and the enhanced $\overline{w*}$ convergence is the reason of the downward-propagating positive temperature anomaly (Andrews et al., 1987), which in turn enhances the high-latitude TIL once the signal reaches the lowermost stratosphere in winter or spring.

- Baroclinic waves and their embedded cyclones-anticyclones can enhance the TIL by tropopause lifting and cooling, and also warming above the tropopause in anticyclones from vertical wind convergence (from model experiments by Wirth (2003, 2004); Wirth and Szabo (2007); Son and Polvani (2007)). These synoptic-scale dynamics partly explain the seasonality and latitude-dependence of the extratropical TIL strength. Also, the baroclinic life-cycle experiment by Erler and Wirth (2011)

showed the importance of baroclinic wave breaking events in enhancing the TIL. However, so far there is only observational evidence of the role of synoptic-scale dynamics in the cyclone-anticyclone modulation of the (weaker-stronger) TIL (Randel et al., 2007; Randel and Wu, 2010; Pilch Kedzierski et al., 2015).

- And finally, small-scale inertia-gravity waves (IGW) also play a role in enhancing the TIL.

Kunkel et al. (2014) showed transient TIL modulation and enhancement from the presence of IGW's in a baroclinic life-cycle experiment, and proposed that this could persistently enhance/maintain the TIL via wave-mean flow interaction. This was confirmed in the study by Zhang et al. (2015), who showed that the strong wind shear found within the TIL lead to IGW breaking, downward heat flux and tropopause cooling (from a single US high-resolution radiosonde station).

Note that the majority of these studies focused on permanent and irreversible processes leading to TIL enhancement. The goal of our study is two-fold: 1) quantify how much of the TIL strength in the extratropics comes from its transient modulation by planetary to synoptic-scale waves, and 2) by subtracting the wave signal, to identify other processes that enhance the TIL and thereby make an observational confirmation of their relative contribution. Only modelling studies have looked at

these processes thus far (reanalysis in the case of a major SSW in Wargan and Coy (2016)), or their observation is very sparse (one single US high-resolution radiosonde station in the IGW study by Zhang et al. (2015)). Therefore it is of interest to put the roles of the different TIL enhancing processes, as enumerated above, into context by using large amounts of high-resolution global GPS-RO observations from the COSMIC mission (Anthes et al., 2008), which is the primary source of

high-resolution observations of the temperature structure near the tropopause. It has to be pointed out that our focus is oriented on knowing the total signal of the extratropical (planetary to synoptic-scale) waves, rather than separating every possible wave type at each given time, which is not possible in practice given the highly variable background wind regimes in the extratropics (see section 2 for more details).

We extract the extratropical wave signal by wavenumber-frequency domain filtering of gridded GPS-RO data. Our method is similar to that of Pilch Kedzierski et al. (2016), who quantified the role of the different equatorial wave types in modulating and enhancing the tropical TIL. Compared to the equatorial wave filtering, the method in this study is adapted and simplified to account for the distinct wave spectrum and the highly varying wind regimes that are found in the extratropics. We

explain how this is done, justifying the filter settings and disclosing the wave modulation mechanism in detail, in section 2; and we provide a proof of concept in section 3, showing that our method is successful in representing the extratropical waves and their TIL modulation. In section 4 we quantify the average signal of the extratropical waves as well as the remaining TIL with this signal removed, for all seasons in the mid- and high-latitudes of the Northern and Southern Hemispheres (NH and

SH). We discuss our results and relate the wave modulation mechanism to previous studies in section 5. The main findings are summarized in section 6.

## 2   Data and Methods

### 2.1   Datasets

We analyze GPS-RO temperature profiles measured by the COSMIC satellite mission (Anthes et al.,

2008). The effective physical resolution of GPS-RO retrievals is normally ∼1km, improving towards the order of 100m in regions where the stratification of the atmosphere changes, such as the tropopause and the top of the boundary layer, where high vertical resolution is most needed (Kursinski et al., 1997). The COSMIC dataset is provided (interpolated) on a regular vertical grid with 100m spacing, from the surface up to 40km altitude. The vertical resolution and height range of the

COSMIC GPS-RO temperature profiles are close to those of radiosonde data, but with the advantages of global coverage, high sampling density of ∼2000 profiles/day, and weather-independence. Also, the accuracy of GPS-RO profiles is even higher than that of radiosondes (Anthes et al., 2008; WMO, 1996). With the temperature profiles, vertical profiles of static stability are calculated as the Brunt-Väisälä frequency squared ($N^2$ [$s^{-2}$]):

$N^2 = (g/\Theta) \cdot (\partial\Theta/\partial z)$

where g is the gravitational acceleration, $\Theta$ the potential temperature, z the vertical dimension and $\partial$ its partial derivative. The tropopause height ($TP_z$) was calculated using the World Meteorological Organization lapse-rate tropopause criterion (WMO, 1957). Profiles with unphysical temperatures or $N^2$ values (temperature <-150°C, >150°C or $N^2$ >100×$10^{-4}s^{-2}$) or those where the tropopause

was not found were excluded (< 1‰).

To extract the extratropical wave signal from the GPS-RO dataset, we first grid the profiles (see subsection 2.2) and then apply wavenumber-frequency domain filters which require data on a regular longitude-time spacing (see subsection 2.3).

To complement the GPS-RO observations, we also use daily-mean vertical profiles of zonal winds from the ERA-Interim reanalysis (Dee et al., 2011) for the years 2007-2013, which provide information about seasonal wind changes or polar vortex disturbances.

## 2.2   Gridding of GPS-RO profiles

The COSMIC GPS-RO temperature profiles at certain latitude bands are gridded daily, between 2007-2013, on a regular 10° longitude grid. The latitude bands chosen for this study are 40°N and 40°S to represent the mid-latitudes, and 80°N and 80°S for polar latitudes. These latitude bands were selected because they show the seasonal cycle of the tropopause and TIL best for mid and high latitudes. Throughout section 4, results for the latitudes 40° and 80° will be disclosed for both Hemispheres. The same analyses were performed for the latitude bands in between, 50°-60°-70°, which have an intermediate behavior in each case.

At each grid point, the profiles of that day within +-5° latitude and +-5° longitude are selected to calculate a tropopause-based weighted average temperature profile and the corresponding $N^2$ vertical profile:

$$T_{grid}(\lambda, Z_{TP}, t) = \sum_i w_i T_i(\lambda, Z_{TP}, t) / \sum_i w_i$$

$$N^2_{grid}(\lambda, Z_{TP}, t) = \sum_i w_i N^2_i(\lambda, Z_{TP}, t) / \sum_i w_i$$

where $\lambda$ is longitude, $Z_{TP}$ is the height relative to the tropopause and t is time. The weight $w_i$ is a Gaussian function that depends on the distance of the GPS-RO profile from the grid center, taking longitude, latitude and time (distance from 12UTC):

$w_i = exp(-[(D_x/5)^2 + (D_y/5)^2 + (D_t/12)^2])$ , where D are the distances in °longitude (x subscript), °latitude (y) and hours (t). The maximum distance allowed from the grid point in each dimension is: 5° longitude, 5° latitude, and 12 hours from 12UTC, respectively.

The gridded tropopause height ($\lambda$,t) is calculated with the same weighting of all profiles' tropopauses. The gridded temperature and $N^2$ profiles are shifted, as the last step, from the tropopause-based vertical scale onto a ground-based vertical scale from 5km to 35km altitude, obtaining a longitude-height array for each day of 2007-2013.

Most often 2-3 profiles are selected for averaging at a grid point with these settings, although one GPS-RO profile is sufficient to estimate a grid point. However, in 14.8% of the cases the algorithm does not find any profile. To fill in the gaps, the longitude range to select the profiles is incremented

to +-10° instead of +-5°, and the latitude range is also incremented to +-7.5° instead of +-5°, which then leaves a 1.8% of empty grid-points. For this minority, profiles are selected within +-1day and +-15° longitude (and same latitude settings). In all cases the weighting function remains the same. The remaining data gaps (0.06%) are filled by averaging neighboring grid points (first +-1 in longitude, then +-1 in time). These exceptions are for a very small portion of the gridded data, and therefore do not affect the retrieved wave signatures after filtering (the percentages showed in this paragraph are for the Northern Hemisphere, and they are similar in the SH).

This gridding method is very similar to the one used by Pilch Kedzierski et al. (2016) at the equator, developed after Randel and Wu (2005). The higher GPS-RO density in the extratropics allowed for a narrower latitude range to select profiles around a latitude band in order to minimize meridional smoothing of the extratropical TIL properties. The lower bound of the vertical scale is set to 5km to account for the lower extratropical tropopause heights compared to the equator.

Once gridded, for each latitude band, we end up with a daily longitude-time array of temperature and $N^2$ for each level between 5-35km with 100m vertical spacing. With a daily temporal resolution, this dataset resolves waves with periods of 2 days or longer, or frequencies of 0.5 cycles per day (cpd) or lower, which is enough to capture most of the planetary and synoptic-scale extratropical waves. Note that the inertial frequency in the extratropics ranges from ∼1.3cpd at 40° latitude to 2cpd at the poles, so inertia-gravity waves (IGW), with frequencies between the inertial and the much higher buoyancy frequency, cannot be resolved with these settings (Andrews et al., 1987). However, by extracting the combined planetary and synoptic-scale wave signal and subtracting it later, we can indirectly infer how important is the role of IGWs or other processes in enhancing the extratropical TIL.

## 2.3 Wavenumber-Frequency Domain Filtering

With the longitude-height-time array of gridded temperature and $N^2$ profiles obtained in the previous subsection 2.2, we make use of the freely available 'kf-filter' NCAR Command Language (NCL) function (Schreck, 2009) to filter in the wavenumber-frequency domain. For each vertical level (from 5km to 35km height with 100m vertical spacing), we retrieve a longitude-time array which is detrended, tapered in time and space-time bandpass filtered with a two-dimensional Fast Fourier Transform. This methodology is analogous to that of Pilch Kedzierski et al. (2016), who filtered gridded equatorial GPS-RO data in certain regions of the wavenumber-frequency domain, following the dispersion curves of the different equatorial wave types which have clear spectral signatures in those wavenumber-frequency domains (Wheeler and Kiladis, 1999). In the remainder of this subsection we will explain how the filter bounds in the wavenumber-frequency domain were chosen in this study to adapt the method of Pilch Kedzierski et al. (2016) to the extratropics.

The extratropical wave modes differ greatly from those at the equator: they are not equatorially trapped and can propagate in any direction, the Coriolis parameter has to be taken into account,

and the zonal wind regimes in the extratropics are much more variable than the equatorial ones. At sub-seasonal time-scales, the Northern Annular Mode (NAM) can alter the strength of the prevailing westerlies in the extratropical troposphere and lowermost stratosphere; there is a strong seasonal cycle in the stratosphere with very strong westerlies (several tens of m/s) within the winter polar vortex, changing to easterlies (-10 to -20 m/s) in summer; and the polar vortex is disrupted very quickly during major SSWs. These modes of variability of the zonal winds in the extratropics have higher amplitude and frequency than any wind regime shifts found at the equator, which has a relatively weak seasonal cycle, and the quasi-biennial oscillation (QBO) in the stratosphere still has less amplitude and lower frequency compared to the polar winter vortex - summer anticyclone. Background winds affect the propagation of waves, by doppler-shifting their dispersion relations or even precluding their propagation, therefore it is of special importance to take them into account in the extratropics.

We make a comparison of the dispersion curves of the extratropical Rossby wave under different background zonal wind regimes in Figure 1. The Rossby wave dispersion relation is defined as the most common form of large-scale wave disturbance found in the extratropics: a planetary wave forced from the troposphere, and propagating vertically and zonally in a quasi-geostrophic flow. Assuming $N^2$ and background mean zonal winds ($\bar{U}$) to be constant, and neglecting meridional propagation for simplicity, the following shallow-water dispersion relation can be obtained (following Andrews et al. (1987)):

$$w = s\bar{U} - s\beta[s^2 + f^2/gh]^{-1}$$

where $w$ is the frequency, $s$ is the zonal wavenumber, $f$ the Coriolis parameter, $\beta$ its meridional derivative at a certain latitude (the Beta-plane approximation), $g$ the gravity acceleration and $h$ the equivalent depth. Since we neglect meridional propagation, the meridional wavenumber is set to zero so it is absent in this formula (compared to Andrews et al. (1987)). The term ($s\bar{U}$) accounts for the doppler-shifting of the dispersion relation by the background zonal winds; and the ($f^2/gh$) term is the vertical wavenumber describing the vertical propagation of the wave in terms of equivalent depth.

In Figure 1, we show how the dispersion curve of a Rossby wave changes depending on its equivalent depth (different line types), and the background zonal mean winds (zero winds black, blue for easterlies, red for westerlies, see specific arrows outside the diagram). Note that each dispersion curve is not valid for the entire year for the Rossby wave. For example, a Rossby wave in winter would propagate following the red (10m/s) dispersion relation in the lowermost stratosphere (i.e. the bottom of the polar vortex); at higher levels (the core of the polar vortex) stronger westerlies are found and the same Rossby wave would propagate following the red (40m/s) curve. At some point in spring and autumn, the black curve ($\bar{U}$ = 0 m/s) is theoretically valid, and so is the blue curve (-10m/s) in summer, with easterlies throughout the stratosphere.

The dispersion relations in Fig. 1 show the difficulty of defining a wave type in the extratropics: the dispersion curve for a given wave type (idealized Rossby wave in this case) can be in basically every possible place within the wavenumber-frequency domain, depending on the background winds. In reality, there is more complexity when meridional propagation and vertical/horizontal wind shear are taken into account for the wave's dispersion relation, which would further increase the difficulty of defining it in a certain region of the wavenumber-frequency domain. Therefore, it is impossible to define one filter to extract Rossby waves that is valid for the entire time period (2007-2013) and that could be used at all levels between 5 and 35km altitude.

We overcome this difficulty by simplifying the method of Pilch Kedzierski et al. (2016). Instead of defining certain dispersion curves, we use wide boxes in the wavenumber-frequency domain, only differentiating eastward-westward propagating oscillations with respect to the ground and their periods (faster 2-4 day waves; slower 4-25 day waves; and 30-96 day or quasi-stationary waves), which are displayed as the six grey boxes in Fig. 1. We also define a seventh filter for wavenumber zero (s = 0, brown box in the middle of the diagram in Fig. 1) for completeness, which represents zonally-symmetric, annular oscillations in time (any possible pole-trapped wave mode). This way, the Rossby waves will be captured by one or another filter, independently of the background zonal winds -together with any other extratropical wave mode that oscillates in those wavenumber-frequency domains. We stress that our gridded data set resolves waves of periods of 2 days or longer, which excludes IGWs and GWs (see subsection 2.2).

With this method we prioritize knowing the total effect of planetary and synoptic-scale extratropical waves on the TIL, at the cost of not differentiating baroclinic, barotropic, standing, travelling (etcetera) wave modes. We find this to be a fair compromise, since our study targets TIL modulation and enhancement by extratropical waves (successfully, see sections 3 and 4), and not to disclose particular properties of these waves.

If waves are present, their filtered signal stands out from the (unavoidable to filter) background noise, which appears as a continuum of low-amplitude fluctuations (Wheeler and Kiladis, 1999), and the transient modulation of the tropopause and the TIL by these waves will be captured since these oscillations are resolved by the gridded dataset (see subsection 2.2). When waves are not present, only the low-amplitude background noise will appear in the filtered signal, and the observed tropopause heights will not be modulated by a non-existing anomaly, therefore having no effect on the tropopause-based means. In this case, the use of broad boxes in the wavenumber-frequency domain for filtering is not a drawback.

The wave modulation mechanism is explained in detail in the next subsection 2.4.

## 2.4 The Wave Modulation Mechanism

As mentioned in the previous subsection 2.3, the extratropical wave anomalies are filtered from a longitude-time array: each vertical level of the gridded GPS-RO profiles is filtered independently,

and then stored together in a longitude-height-time array of wave anomalies. Therefore, for a given latitude band, we end up with arrays of gridded GPS-RO profiles (temperature and $N^2$), and the corresponding anomalies (also of temperature and $N^2$) from the seven wave filter bounds defined in Fig. 1, all gridded with 10° longitude, 100m height and 1-day spacing.

When one specific day is selected from these arrays, a longitude-height snapshot of the wave anomalies is obtained. Extratropical waves have vertical tilts in their temperature structures, and if the anomalies are large, they can effectively modulate tropopause height as explained next.

Figure 2a shows idealized temperature anomalies associated with an atmospheric wave with west-ward vertical tilt, as a longitude-height snapshot of the positive/negative (red/blue dashed contours) temperature anomaly structures. The anomalies sketched in Fig. 2a would correspond to that of an upward-propagating Rossby wave (Andrews et al., 1987) which is present around the extratropical tropopause. Any tropospheric or stratospheric anomalies are omitted for simplicity in Fig. 2a, since they are irrelevant for the transient modulation of the TIL. The local temperature profile would be the result of superimposing the local anomalies to the ground-based zonal mean temperature profile, as depicted in Figs. 2b and 2c.

The thermal tropopause is defined as the point where the lapse-rate changes from typical tropo-spheric values (∼7K/km colder with height) to less than 2K/km for more than 2km (WMO, 1957). Therefore, if the wave temperature anomalies are large enough, they could effectively change the point where this criterion is met by deviating the local lapse-rate near the tropopause from that of the zonal-mean profile. Fig. 2b shows a case where the tropopause is lowered due to the wave anomalies, and Fig. 2c another where the tropopause is lifted. The tropopause would tend to be placed above negative stratification anomalies, and/or below positive stratification anomalies, which is most often centered at the strongest negative temperature anomaly as in both idealized cases in Figs. 2b and 2c.

As explained in subsection 2.3, these temperature anomalies are not only deviations from the ground-based zonal mean profile, but they are also part of a space-time harmonic: they need to be wave-like and to travel in the wavenumber-frequency domain by definition. Local anomalies present in the gridded GPS-RO temperature profiles that are not part of a planetary or synoptic-scale wave (but the result of other processes) remain unfiltered and are not included within the analyzed wave anomalies.

Note that the tropopause modulation by atmospheric waves sketched in Fig. 2 is highly idealized: usually waves with different wavenumbers, frequencies and vertical tilts can be present simultane-ously, making the actual observed tropopause modulation more complex than in Fig. 2. As explained next, our filtering method applies to any vertical tilt.

We highlight that the wavenumber-frequency domain filters are applied to longitude-time arrays from each vertical level independently (see subsection 2.3): no filtering is done in the vertical direction. Therefore, if an atmospheric wave is present, it will be captured by our filters regardless of its vertical tilt, as long as at a given level it can be represented by a space(longitude)-time harmonic. The

vertical tilt of the wave appears when the filtered anomalies of each vertical level are piled together (see section 3 and Figure 3).

In Fig. 2 we presented idealized upward-propagating Rossby wave temperature anomalies to sketch the wave modulation mechanism at the tropopause region because this wave type is the most common in the extratropics. As our filtering method, the wave modulation mechanism also applies to waves with any vertical tilt. One may just mirror Fig. 2a in the longitude direction to get the wave modulation by a wave with eastward vertical tilt (e.g. a downward-propagating Rossby wave).

For waves whose vertical tilt is flat (axisymmetric vortices) or vertical (barotropic wave modes), the anomalies do not form a dipole in the vertical direction, but they do so in the longitude direction: one may take just the upper (or the bottom) pair of anomalies from Fig. 2b and 2c, and still modulate tropopause height from these wave anomalies within a certain longitude range.

A tropopause-based mean of the anomalies sketched in Fig. 2a would yield a dipole of cold anoma-
lies centered at the tropopause, warm anomalies above, and a net TIL enhancement just from the presence of the wave in the tropopause region. Note that the ground-based zonal average temperature profile and the zonal mean tropopause height are unaffected by the wave anomalies in Fig. 2a: only the tropopause-based zonal mean profile is affected. To leave an imprint in the tropopause-based mean, the wave modulation mechanism does not need to be present over the whole longitude
range, but only a significant part of it. Sections 3 and 4 will show that this happens constantly due to the near-ubiquitous presence of waves (of any type) near the tropopause region.

We expect that our filtered wave anomalies will modulate the tropopause in a similar fashion as sketched in Fig. 2a for temperature. Once a wave is away from the tropopause region or dissipated, no filtered signal will affect the TIL, therefore permanent effects from wave-mean flow interaction
(wave dissipation or breaking) are not quantified by our method.

Next in section 3, we show examples of the filtered signals and the tropopause adjustment to the extratropical waves, in order to proof the validity of our method to study the extratropical TIL wave modulation.

## 3   Proof of Concept

Figure 3 shows snapshots of the longitude-height $N^2$ anomalies (colors) filtered in the wavenumber-frequency domains defined in Fig. 1 (see previous subsection 2.3), together with the tropopause height (black line), for the 50°N latitude band. Each snapshot is selected for a different winter date, in order to portray cases when there is a clear modulation of the tropopause's zonal structures from the extratropical wave anomalies. We do not include the s=0 filter since it lacks zonal structures
by definition. In Fig. 3 the eastward and westward propagation refers to the movement of the wave relative to the ground as defined in subsection 2.3.

In Fig. 3 there are several analogies to the tropopause and TIL modulation by equatorial waves described in the study by Pilch Kedzierski et al. (2016). In our extratropical case, we also show tropopause modulation by the wave anomalies when their amplitude is large, with predominant positive $N^2$ anomalies above the tropopause, and negative $N^2$ anomalies below, which correspond to the temperature anomalies that modulate tropopause height as sketched in Fig. 2. The modulation by extratropical waves is especially clear in Fig. 3c for eastward-propagating waves with periods of 4-25 days: strong positive $N^2$ anomalies are detected right above the tropopause between -180°E and -25°E, while negative $N^2$ are located below the tropopause between 25°E and 180°E. Similarly, in Fig. 3d (westward-propagating 4-to-25-day waves) tropopause height follows the positive $N^2$ anomalies between -75°E and 180°E. Sometimes positive $N^2$ anomalies can be located below the tropopause and viceversa, as in the cases shown in Fig. 3 a, b, e and f; but the zonal mean is still dominated by positive $N^2$ anomalies right above the tropopause. Note that different wave types can be present simultaneously within a single spectral region defined in Fig. 1, and thus shape together the $N^2$ anomalies and the tropopause zonal structures appearing in the snapshots in Fig. 3.

It can also be observed in all cases in Fig. 3 that there is a relative maximum of wave activity around the tropopause regardless of the amount of wave activity in the stratosphere, as measured by $N^2$ anomalies. This is in line with the findings of Pilch Kedzierski et al. (2016) who reported wave amplification near the tropopause for every equatorial wave type.

The westward tilt of the Rossby waves can be discernible in many cases (Fig. 3 a-d) throughout the stratosphere: most clearly in the intermediate periods of 4-25 days which are the most common for travelling Rossby waves, but also visible sometimes in the 30-96 day periods, indicating the presence of quasi-stationary Rossby waves. This is a good indicator that these waves are properly captured by our filters, which are used at each vertical level independently: the vertical structure of the waves is obtained without filtering in the vertical direction. Note that in Fig. 3, planetary and synoptic-scale waves are all superimposed, so the overall appearance is increasingly patchy when short and fast waves are present, which is why Fig. 3 e and f show these structures the most.

The wavenumber-frequency domain filtering method used for extratropical waves (see subsection 2.3 and Fig. 1), although simplified compared to Pilch Kedzierski et al. (2016) and unable to differentiate particular wave types, is able to capture the overall planetary and synoptic-scale extratropical wave signal and how it modulates the tropopause and the TIL as shown in Fig. 3. Although Fig. 3 shows very clear cases for each filtered band, in the seasons where the wave signature on the TIL is strongest (mid-latitude winter and polar summer, see section 4) the modulation of the tropopause height zonal structures by the wave anomalies can be as evident almost every day for the band corresponding to Fig. 3c, with one or more other bands also showing significant zonal ranges of tropopause modulation at the same time.

The examples shown in Fig. 3 are for the 50°N latitude band in winter, but similar conclusions can be drawn from any extratropical latitude or season. The tropopause-based, seasonal mean of the

extratropical wave signal will show an overall TIL enhancement, which is quantified next in section
4 for mid and high latitudes, winter and summer, Northern and Southern Hemisphere.

## 4 Wave Modulation of the Extratropical Tropopause Inversion Layer

Figure 3 in section 3 showed the tropopause adjustment to the horizontal and vertical structure
of the filtered extratropical wave anomalies, with positive $N^2$ anomalies generally placed above
the tropopause, and negative $N^2$ anomalies below. Therefore, a tropopause-based average of the
wave signals should give a net TIL enhancement. We perform the same analysis with the filtered
temperature anomalies, which we expect to show a net cold anomaly at the tropopause and a warm
anomaly aloft (the dipole needed to enhance $N^2$ right above the tropopause).

   Also, by subtracting the extratropical wave signal from the gridded GPS-RO data, it is possible
to show the remaining TIL that is caused by mechanisms other than the extratropical wave modula-
tion. We will present the daily evolution of the vertical tropopause-based $N^2$ profile, comparing the
observed $N^2$ vertical structure to the one without the extratropical wave signal, which should show
a weaker TIL.

   Note that the wavenumber-frequency domain filters are not able to extract the wave anomalies at
the beginning and end of the 2007-2013 time-period of our study. The longest period filtered is 96
395    days (see section 2, Fig. 1), therefore data from the first 100 days of 2007 and the last 100 days of
2013 are not used for any figures of this section, in order to make sure that there is no signal missing.

   Throughout this section, we will present the two kinds of analysis explained in the previous para-
graphs: seasonal, tropopause-based averages of the extratropical wave temperature and $N^2$ signals;
and the time evolution of observed $N^2$ zonal-mean profiles, with and without the extratropical wave
signal. Both analyses will be presented for mid-latitudes (40°N, subsection 4.1) and polar latitudes
(80°N, subsection 4.2), first in the Northern Hemisphere, and then the exact same methodology is
applied to the same latitude bands in the Southern Hemisphere (80°S in subsection 4.3; 40°S in
Appendix A).

### 4.1 Northern Hemisphere Mid-latitudes

Figure 4 shows the seasonally averaged signature of the different extratropical wave spectral re-
gions (defined in subsection 2.3, figure 1) at 40°N, as their mean anomaly in the tropopause-based
zonal-mean vertical profiles of temperature (left column) and $N^2$ (right column). All the defined
extratropical wave spectral regions show a mean cold anomaly maximizing at the tropopause (Fig. 4
a and c), and a $N^2$ increase directly above the tropopause (Fig. 4 b and d), thereby producing a net
TIL enhancement. This is also in line with the findings by Pilch Kedzierski et al. (2016), who found
the same effect of all equatorial wave types on the tropical TIL, only varying in the amplitude of the
mean wave signature. The mean wave signatures in Figure 4 show that extratropical waves enhance

the TIL in a very similar manner by tropopause adjustment to the wave signal and the resulting cold anomaly at the tropopause (and a warm anomaly above to a lesser degree).

In Fig. 4 (a-d), the strongest signal belongs to eastward-propagating waves with periods of 4 to 25 days (red lines), in both winter (top row) and summer (bottom row). Baroclinic Rossby waves, the most common wave type occuring at mid-latitudes (with prevailing westerlies during all year at near-tropopause level, therefore their eastward propagation respect to the ground), fit within this broadly defined wavenumber-frequency domain. This also explains why the extratropical wave sig-
nal is stronger in winter at mid-latitudes, since the mid-latitude jet strength and the baroclinic wave activity both peak there during winter. We also note that quasi-stationary waves (periods of 30-96 days, black and dashed magenta lines) and the s=0 wave (grey line) play a minor role in enhancing the TIL.

The total extratropical wave signal (Fig. 4 e and f) at 40°N is a mean cold anomaly at the tropopause of $\sim$3.5K and a TIL enhancement of $\sim$1.6$\times$10$^{-4}s^{-2}$ in winter (red line). In sum-
mer (black line) the modulation is weaker: tropopause-centered mean cold anomaly of $\sim$2.3K and $\sim$1.1$\times$10$^{-4}s^{-2}$ of TIL enhancement. The spring and autumn wave signatures are similar to the results shown here for winter and summer, respectively. The total extratropical wave signature in NH mid-latitude summer has a very similar magnitude compared to the equatorial wave signal obtained
by Pilch Kedzierski et al. (2016).

Figure 5a shows the daily evolution of the tropopause-based $N^2$ profile (2007-2013) at 40°N. In Fig. 5a the TIL is clearly discernible in winter and into spring with higher $N^2$ values right above the tropopause, ranging between 5.5$\times$10$^{-4}s^{-2}$ (orange) and 6.5$\times$10$^{-4}s^{-2}$ (red). Above the TIL in winter, the lowermost stratosphere has a relative minimum in $N^2$ of $\sim$3.5$\times$10$^{-4}s^{-2}$ centered at
15km height, and levels higher than 18km have $N^2$ values of $\sim$5$\times$10$^{-4}s^{-2}$. In summer months the tropopause is higher, and although the TIL and the stratospheric $N^2$ values are separated by a weak relative $N^2$ minimum (white, blueish sometimes), both layers have $N^2$ values of 5-5.5 $\times$10$^{-4}s^{-2}$ (yellow, orange sometimes). Fig. 5a agrees with previous climatologies of the mid-latitude $N^2$ vertical structure (e.g. Birner (2006); Grise et al. (2010)), while also showing its short-term variability
without time-averages.

Fig. 5b shows the same $N^2$ profile evolution, but with the extratropical wave signal subtracted, therefore displaying the tropopause-based $N^2$ structures without the transient modulation by planetary to synoptic-scale extratropical waves. Therefore, any TIL observed in Fig. 5b should be caused by other processes. It can be observed that the TIL in Fig. 5b is almost completely gone: $N^2$ right
above the tropopause is always lower than the stratospheric values above 18km. However, in winter and spring a very weak relative maximum of $N^2$ can be observed above the tropopause (4-4.5 $\times$10$^{-4}s^{-2}$, white and light-blue colors compared to the $N^2$ minimum of 3.5$\times$10$^{-4}s^{-2}$ at 15km height), occasionally reaching $N^2$ values close to 5$\times$10$^{-4}s^{-2}$ (sparse light-yellow spots) in late

winter and spring. In summer, there is no relative $N^2$ maximum above the tropopause at all in Fig.
5b.

The conclusion that Fig. 5b gives is that most of the mid-latitude TIL is explained by the transient modulation by planetary to synoptic-scale extratropical waves. Other possible sources of TIL enhancement in the extratropics like IGW modulation, wave-mean flow interactions of any wave type, residual circulation or radiative effects; all together they play a minor role in forming the zonal-mean TIL structure. The TIL enhancement by IGWs (Zhang et al., 2015) can be of importance locally in space and time, but its contribution to the zonal-mean TIL (even if it explained all the structures in Fig. 5b) would be less than the effect of the filtered planetary and synoptic-scale extratropical waves.

Separating the different extratropical wave types and their contribution to TIL enhancement is beyond the scope of this study, but of interest for future research. Two questions arise from our results:

1) Which wave type is dominant? Figs. 3 and 4 point towards the baroclinic Rossby wave, the most common and strongest wave type occurring in the extratropics, and we find the biggest signals in the broad wavenumber-frequency domain that would include this wave type, but this still needs robust confirmation. Our current method would need significant refinement to separate the baroclinic Rossby wave from other wave modes present in the extratropics.

2) Which is the process that leads to the amplification of extratropical waves next to the tropopause level? This is visible in Fig. 3 for all the wave spectral regions defined in subsection 2.3, and analogous to the near-tropopause amplification of all equatorial wave types in the tropics observed by Pilch Kedzierski et al. (2016). It would be of high interest to know whether this amplification follows Linear Wave Theory (Andrews et al., 1987) or not.

The conclusions from this subsection for 40°N also apply to the Southern Hemisphere. The equivalent analyses for 40°S can be found in Appendix A.

### 4.2 Northern Hemisphere Polar latitudes

We proceed to apply the same analysis from the previous subsection 4.1 to polar latitudes. Figure 6 shows the seasonal average signature of the different extratropical wave spectral regions (defined in subsection 2.3, Fig. 1) at 80°N, as their mean anomaly in the tropopause-based zonal-mean vertical profiles of temperature (left column) and $N^2$ (right column). As in mid-latitudes (subsection 4.1), all the defined extratropical wave spectral regions show a mean cold anomaly maximized at the tropopause (Fig. 6 a and c), and a $N^2$ increase right above the tropopause (Fig. 6 b and d). However, at polar latitudes the seasonality of the extratropical wave forcing is inverted compared to mid-latitudes. The total wave signatures in Fig. 6 e and f are weaker in winter (red line), with a tropopause mean cold anomaly of $\sim$2.6K and a TIL enhancement of $\sim$1.1$\times10^{-4}s^{-2}$, similar to that found in mid-latitude summer. In summer (black line), there is a total tropopause mean cold anomaly of $\sim$3.9K and a TIL enhancement of $\sim$1.9$\times10^{-4}s^{-2}$, similar to that found in mid-latitude winter. The

spring and autumn wave signatures at NH polar latitudes are both in between the values for winter and summer shown here.

     Note that in Fig. 6 (a-d) the eastward-propagating 4-25 day band (red line) is no longer as dominant as in Fig. 4. This can be explained by the fact that zonal mean westerly winds are weaker at polar latitudes, therefore the wave spectrum does not get Doppler-shifted as much as at mid-latitudes,

and more waves are observed to be westward-propagating with respect to the ground. As to why the extratropical wave signature is stronger in polar summer, as opposed to mid-latitudes (stronger in winter), we explain the inverted seasonalities by the position of the jet stream and the baroclinic wave activity, which migrate polewards in summer while being at the mid-latitudes in winter. However, it is still surprising that the total extratropical wave signature on the TIL region is of the same

magnitude at mid-latitudes and polar latitudes, despite the opposed seasonal cycles as explained above. Since the meridional temperature gradients and the jet stream are weaker in summer, one would expect the extratropical wave signature on the TIL to follow this tendency. It is possible that extratropical waves at polar regions get amplified near the tropopause and reach the same amplitude as at mid-latitudes (at opposing seasons), but as noted in subsection 4.1 this amplifying mechanism

near the tropopause needs further research.

     Figure 7a shows the daily evolution of the tropopause-based $N^2$ profile (2007-2013) at 80°N. There is a distinct TIL throughout the year, with $N^2$ values right above the tropopause of $\sim 5 \times 10^{-4} s^{-2}$ in winter (white to yellow colors) and between 7-8 $\times 10^{-4} s^{-2}$ in summer (brown and black sometimes). Stratospheric $N^2$ values are around $4 \times 10^{-4} s^{-2}$ (dark and light blue) at levels within 14-

26km height, with increasing $N^2$ at higher levels. The $N^2$ structures showed in Fig. 7a (as in Fig. 5a) agree with previous climatologies of the high-latitudes $N^2$ vertical structure (Birner, 2006; Grise et al., 2010), and the daily temporal resolution shows the high variability associated with sudden stratospheric warmings (SSWs) in the stratosphere. Higher $N^2$ values in the stratosphere are observed during SSWs, with positive $N^2$ anomalies propagating downward and reaching the TIL

region. The SSWs signals at particular events will be discussed next, since they will be easier to differentiate once the extratropical wave signal is removed in Fig. 7b.

     Fig. 7b shows the $N^2$ profile evolution without the extratropical wave signal, displaying the tropopause-based $N^2$ structures caused by other processes. The TIL in Fig. 7b is significantly weakened without the extratropical wave modulation: in winter it almost disappears, but in summer the

TIL is still distinct (5-6 $\times 10^{-4} s^{-2}$, yellow and orange) from the background stratospheric $N^2$ structure (blue). The extratropical wave modulation explains an important part of the TIL's $N^2$ structure in polar latitudes (a similar amount of $N^2$ enhancement as in mid-latitudes, with inverted seasonality, Fig. 6), but other sources of TIL enhancement are also present as it can be observed in Fig. 7b (unlike in Fig. 5b for mid-latitudes, where almost no TIL is visible without the extratropical wave

signal). Most notably, the removal of the extratropical wave signal makes the time evolution of the vertical $N^2$ structures in Fig. 7b much smoother compared to Fig. 7a, and allows a clearer appear-

ance of the downward-propagating signal from SSWs and how it affects the tropopause region. In Fig. 7b, major SSW events are marked with black arrows (2008, 2009, 2010, 2013), and one minor event is marked with a grey arrow in 2012. The major SSW event from February 2007 is not marked, since the first 100 days of 2007 are cut off for the analyses in this section.

During major SSWs, the residual circulation is accelerated, and the convergence of its vertical component ($\overline{w*}$) forces a positive temperature anomaly that propagates downward into the lowermost stratosphere (Andrews et al., 1987). In the study by Wargan and Coy (2016) it was shown that $\overline{w*}$ convergence is associated with a downward-propagating positive $N^2$ anomaly as well, that enhances the TIL once the SSW signal reaches the tropopause region. Wargan and Coy (2016) calculated a $\sim 1.5 \times 10^{-4} s^{-2}$ increase of the zonal-mean $N^2$ maximum above the tropopause due to the 2009 major SSW, and slightly lower $N^2$ increases in other SSW cases. In Fig. 7b, it can be observed that $N^2$ right above the tropopause in early 2009 increases from $\sim 4 \times 10^{-4} s^{-2}$ (blue) before the SSW, up to $\sim 5.5 \times 10^{-4} s^{-2}$ (orange) after the SSW. Also, a positive $N^2$ anomaly from the 2009 major SSW can be seen in Fig. 7b propagating downwards throughout the stratosphere (white and yellow instead of blue), and the TIL enhancement coincides with the time when this downward-propagating anomaly reaches the tropopause region, as well as a marked decrease in the zonal-mean tropopause height. This perfectly fits the findings of Wargan and Coy (2016). In Fig. 7b the same can be observed in the major SSW cases of 2008, 2010 and 2013, although the $N^2$ anomalies are slightly lower than in 2009 which was an exceptionally strong event.

Interestingly, in Fig. 7b we observe the downward-propagating positive $N^2$ anomaly, TIL enhancement and tropopause lowering in a minor SSW in early 2012, and also during the final warmings of 2011 and 2013. The coherency in time of these signals, and their similarity to the cases described by Wargan and Coy (2016) suggests that they are also driven by an acceleration of the residual circulation (increased $\overline{w*}$ convergence) from the disturbed polar vortex. The 2013 case is quite particular: once the major SSW is finished, the polar vortex recovers, the TIL is no longer enhanced and the tropopause slowly increases its height; but then there is a strong final warming event, another downward-propagating $N^2$ signal, immediate TIL enhancement and a slight lowering of the zonal-mean tropopause. After this, the zonal-mean tropopause gets steadily higher into the summer. In the final warming of 2011 there is an abrupt transition from a strong polar vortex to anticyclonic circulation, and the downward-propagating $N^2$ signal, TIL enhancement and abrupt zonal-mean tropopause lowering is also visible in Fig. 7b. In the case of the minor SSW of 2012, the TIL enhancement and zonal-mean tropopause lowering are also in clear coincidence with the disrupted westerlies.

Figure 7b shows evidence, directly from observations, that the TIL is enhanced due to major SSWs, and also from other polar vortex disturbances: minor SSWs and abrupt final stratospheric warmings. The similarity of our results with Wargan and Coy (2016), who studied major SSWs, in terms of the time evolution of the $N^2$ signal, TIL enhancement and tropopause height; suggests

that accelerated residual circulation (increased $\overline{w*}$ convergence) is the main contributor to TIL en-
hancement during all kinds of polar vortex disturbances, not only major SSWs. This would need
confirmation with a more detailed study of (non-major) polar vortex disturbances and the associated
residual circulation variability.

Fig. 6f showed TIL enhancement of $\sim$1.1$\times$10$^{-4}$$s^{-2}$ by extratropical wave modulation in polar
winter. In Fig. 7b, we show that polar vortex disturbances in general can enhance the TIL in winter
(major and minor SSWs) and spring (final warmings) with a similar magnitude. The remaining
TIL in polar summer in Fig. 7b ($\sim$5.5$\times$10$^{-4}$$s^{-2}$, orange) is not explained by extratropical wave
modulation, nor by residual circulation. The only other known mechanism that could enhance the
TIL in polar summer is water vapor radiative cooling of the tropopause (Randel and Wu, 2010;
Miyazaki et al., 2010b), but this would also require an additional study to be confirmed.

We also note that the meridional advection of the SSW signals in the lowermost stratosphere
could be the cause of the very weak hints of the mid-latitude TIL without the extratropical wave
signal found in Fig. 4b, that mainly appears in late winter and spring and was strongest in 2009,
2010 and 2013, coinciding with major SSW events.

Given that the polar vortex behavior affects the TIL more clearly at polar latitudes, we expect more
differences between the NH and the SH, since the polar vortex in the SH is much less disturbed than
in the NH, and the only major SSW observed in the SH happened in 2002.

### 4.3 Southern Hemisphere Polar latitudes

Figure 8 is the Southern Hemisphere (SH) equivalent of Fig. 6. The total extratropical wave signal
at 80°S (Fig. 8 e and f) is a tropopause mean cold anomaly of $\sim$1.4K and a TIL enhancement of
$\sim$0.9$\times$10$^{-4}$$s^{-2}$ in winter (compared to $\sim$1.4$\times$10$^{-4}$$s^{-2}$ TIL enhancement in the NH in Fig. 6). In
summer there is a tropopause mean cold anomaly of $\sim$2.5K and $\sim$1.3$\times$10$^{-4}$$s^{-2}$ of TIL enhancement
($\sim$1.9$\times$10$^{-4}$$s^{-2}$ in the NH, Fig. 6). The extratropical wave signatures in Fig. 8 show the same
seasonality as in Fig. 6, with stronger (weaker) signals in summer (winter) months, but the overall
magnitude of the extratropical wave forcing at polar latitudes is lower in the SH than in the NH. The
spring and autumn wave signatures at SH polar latitudes are similar to the results shown here for
winter and summer, respectively.

The lower extratropical wave activity and the smaller mean signal at the tropopause near the
South pole is explained by the isolation of the SH polar latitudes: no land-sea contrast or high
mountain ranges in the meridional direction (less wave sources), and a stronger and more stable
polar vortex that does not allow waves to propagate so deep into high latitudes, as opposed to the
NH. The behavior of the extratropical wave forcing in SH polar latitudes (Fig. 8) is similar to the
NH (Fig. 6) but weaker. In subsection 4.2 (NH polar latitudes) it was shown that, after subtracting
the extratropical wave signal, the TIL enhancement from SSWs (major or minor) and final warmings

could be seen clearly. In the 2007-2013 period, no SSW occurred in the SH, so we only aim to see
what is the effect of final warmings.

Figure 9a shows the daily evolution of the tropopause-based $N^2$ profile at 80°S. There is a clear TIL during summer and into autumn, with $N^2$ values of $\sim7\times10^{-4}s^{-2}$ (brown) right above the tropopause. In winter, the TIL is harder to discern, but a weak maximum of $\sim4.5\times10^{-4}s^{-2}$ (white, light yellow) is present above the winter tropopause. The TIL near the South Pole in winter is known to be very weak or absent (Tomikawa et al., 2009; Pilch Kedzierski et al., 2015). Compared to the NH, the SH polar vortex is stronger, less disturbed during winter, and has a longer lifetime: it breaks later in spring, almost into the summer.

Note that in Fig. 9 the tropopause is higher during winter (unlike in Figs. 5 and 7). This seasonal cycle in the high-latitude SH tropopause agrees with previous climatologies from GPS-RO (Son et al., 2011), and is attributed to the very cold and stable polar vortex (Zängl and Hoinka, 2001) and the seasonal cycle in the strength of the Brewer-Dobson circulation (Yulaeva et al., 1994). Also, there is some indeterminacy in the exact height of the thermal tropopause, since the background temperature lapse-rate in SH high-latitudes is close to the WMO lapse-rate tropopause criterion (WMO, 1957) of 2K/km for several kilometers in the upper troposphere during winter. We discuss the downward-propagating signal of the SH polar vortex breakup next.

Fig. 9b shows the $N^2$ profile evolution without the extratropical wave signal, displaying the tropopause-based $N^2$ structures caused by other processes. In summer, the TIL is significantly weaker but clearly present in Fig. 9b. In winter, the TIL cannot be detected without the extratropical wave signal, and the vertical $N^2$ structures are smoother and enable a clearer view of the downward-propagating $N^2$ signal from the SH vortex breakup in late spring. Once the signal reaches the tropopause region, there is an abrupt increase in $N^2$ right above the tropopause, from values of $\sim4\times10^{-4}s^{-2}$ (blue) to $\sim5.5\times10^{-4}s^{-2}$ (yellow-orange), in line with the findings of Wargan and Coy (2016) and our previous subsection 4.2 and Fig. 7b. Even a slight and short-lived relative minimum in tropopause height can be observed with the arrival of the vortex breakup signal. Note that in Fig. 9b, no TIL is discernible until the downward-propagating $N^2$ signal from the SH polar vortex breakup arrives. For example, the contrast between the summers of 2011/12 and 2012/13: in the first summer, the signal reaches the tropopause region right at the beginning of 2012, and the TIL is observed since; whereas in the next summer the polar vortex breaks up early, and the strong TIL is observed more than a month before the beginning of 2013.

Later in the summer, the TIL generally reaches $N^2$ values of $\sim6\times10^{-4}s^{-2}$ every year in Fig. 9b. As in the previous subsection 4.2, we also suggest that the remaining TIL in Fig. 9b in summer is due to water vapor radiative effects which would need further study.

## 5 Discussion

In subsection 2.4 we introduced a new TIL enhancing mechanism, section 3 showed that gridded COSMIC GPS-RO observations are able to capture it, and throughout section 4 we quantified the importance of extratropical wave modulation in explaining the observed mid- and polar latitude TIL strength and variability. Within this section we will discuss how the wave modulation mechanism relates to previously proposed TIL enhancing mechanisms, since it can integrate several of them rather than being a completely separate dynamical process of the tropopause region.

First of all, it is important to clarify what our wave modulation mechanism and the filtered wave anomalies represent exactly. The waves are defined as space-time harmonics of positive-negative (T or $N^2$) anomalies whose sum in the ground-based mean profile is zero, thereby not affecting the background T and $N^2$ profile at all. They represent the adiabatic and transient dynamical effect of the wave: once the wave (as the space-time harmonic) leaves the tropopause region or dissipates, it is not filtered and therefore would no longer influence the tropopause region, which would then be back to its background state. The key detail about this mechanism lies in that, in order to have a significant role in enhancing the TIL, it needs of a constant presence of high-amplitude waves in the extratropical tropopause region. Indeed, GPS-RO observations show this to be the case (see sections 3 and 4).

Another important point to take into account is: what is included within the concept of 'Atmospheric Wave'? Traveling and transient T and $N^2$ anomalies are just a part of the wave that our method is able to quantify, but there are horizontal and vertical motions associated with the wave as well, which can influence water vapor transport and induce the formation of clouds, in turn influencing the local radiative budget and heat fluxes. Also, the waves can interact with the mean flow while breaking and/or dissipating. All of this cannot be observed with GPS-RO measurements, nor quantified with our filtering method.

The benefit of our method is that it separates the transient and adiabatic effect of the extratropical waves (that drive variability around the background T and $N^2$ profiles) from the rest of diabatic processes, associated with the wave or not, whose imprint is present at longer space and time-scales (and can shape the background T and $N^2$ profiles). The latter are still present in Figs. 7b and 9b once the wave signals are subtracted, and agree well with previous literature about the role of the accelerated Brewer-Dobson circulation in enhancing the polar TIL during SSWs (Wargan and Coy, 2016) and water vapor radiative effects in enhancing the polar summer TIL (Randel and Wu, 2010; Miyazaki et al., 2010b).

The wave signature on the tropopause-based, zonal mean T and $N^2$ profiles has a similar magnitude in polar summer and mid-latitude winter (Figs. 4 and 6). At mid-latitudes (Figs. 5b and A2b), the filtered wave signals seem to be responsible for most of the observed TIL. How does this fit to previously proposed TIL enhancing mechanisms? We suggest that, rather than being a completely separate dynamical process, several mechanisms proposed in earlier literature are integrated within

our wave modulation mechanism and the filtered wave signals, particularly when interpreting their transient effects. We will start discussing the dynamical ones and finish with processes related to radiation.

The cyclones and anticyclones embedded within the Rossby wave travel in the same wavenumber-frequency domain. The vertical convergence and isentrope packing happening above the anticyclone and enhancing the TIL (Wirth, 2003, 2004) would therefore be captured by our filters as a positive $N^2$ anomaly (the closer isentropes imply an increased potential temperature gradient). The experiments by Wirth (2003, 2004) involved very idealized axisymmetric vortices, which do not form often in the real atmosphere, but our filtering method is very flexible in capturing the vertical structure of the waves (see subsections 2.3 and 2.4) and the gridded GPS-RO observations capture the lower-higher tropopauses associated with the traveling cyclones-anticyclones, regardless of their symmetry or complexity.

Erler and Wirth (2011) pointed out the role of wave breaking in forming a stronger TIL in their baroclinic life cycle experiments. During the onset of a wave breaking event, the amplitude of the baroclinic wave is at its highest, and its modulation of the tropopause region and the TIL is captured with our method. Our results suggest that this effect is more important than the wave-mean flow interactions associated with the wave breaking, whose more permanent effect should be visible in Figs. 4b and A2b after subtracting the wave signals.

IGWs and GWs can locally modulate $N^2$ near the tropopause and be important for TIL enhancement (Kunkel et al., 2014; Zhang et al., 2015), but our results show that the zonal-mean seasonal TIL is dominated by waves of synoptic and planetary scales.

Regarding radiation, recent modeling studies highlighted the importance of including diabatic processes related to water vapor and clouds, apart from dynamics, in order to explain the mid-latitude TIL (Ferreira et al., 2016; Kunkel et al., 2016). Particularly, the baroclinic life cycle experiments by Kunkel et al. (2016) that included diabatic effects from water vapor and clouds showed a faster deepening of the wave and strengthening of the TIL, along with an increase in the cross-tropopause gradient of water vapor which would in turn increase radiative cooling of the tropopause.

We propose that radiation from water vapor and clouds could enhance the TIL in two different ways: **1)** directly influencing the UTLS background T and $N^2$ profiles, as probably is the case for the polar summer TIL which is supported by previous literature (Randel and Wu, 2010; Miyazaki et al., 2010b), and Figs. 7b and 9b; and **2)** by also setting up a radiative-dynamical feedback that amplifies atmospheric waves near the tropopause which would explain why our filtered wave anomalies are maximized near the tropopause (Fig. 3) and the absence of a TIL in Figs. 5b and A2b after removing the wave signals, while also fitting with the results by Ferreira et al. (2016) and Kunkel et al. (2016). Radiation would then be of importance for enhancing baroclinic wave growth, among other factors that trigger the wave and its growth near the tropopause. Further research needs to be done in order to verify and disclose the importance of such radiative-dynamical feedback.

In any case, we stress that the TIL enhancement by the wave modulation mechanism (described in subsection 2.4) is ultimately done by the atmospheric wave in a transient and adiabatic way as shown throughout section 4. We suggest that the wave modulation mechanism integrates the effects of different processes postulated in earlier TIL literature, whose exact roles need further assessment. The step forward done by our study is to identify the total influence of planetary and synoptic-scale waves on the observed TIL strength, and its transient and adiabatic nature.

## 6 Concluding Remarks

Our study used a simplified method to extract the total extratropical (planetary to synoptic-scale) wave signal from gridded COSMIC GPS-RO profiles. By tropopause-based zonal averaging of these signals at certain latitude bands, we were able to quantify how much of the extratropical TIL at mid- and polar latitudes is explained by the transient modulation of the tropopause region by the planetary and synoptic-scale waves. By subtracting the extratropical wave signal, we show how much of the TIL is left due to other processes.

We found that extratropical wave modulation explains almost all of the observed TIL strength at mid-latitudes in both hemispheres (Figs. 5 and A2). Therefore we conclude that wave-mean flow interactions, inertia-gravity waves or the residual circulation are of minor importance as TIL enhancing mechanisms there.

At polar regions, extratropical wave modulation is dominant as well in explaining the TIL strength, but there is also a clear signal from SSWs, major and minor, in the Northern Hemisphere, and final warmings in both hemispheres (Figs. 7 and 9). The similarity in the time evolution of all signals from the disturbed polar vortexes in both hemispheres suggests that they are forced by the same mechanism: $\overline{w*}$ convergence from accelerated residual circulation as in the major SSW study by Wargan and Coy (2016).

Also, part of the polar summer TIL strength is not explained by extratropical wave modulation nor by residual circulation. We suggest that the only other known mechanism that could enhance the polar summer TIL is water vapor radiative cooling of the tropopause (Randel and Wu, 2010; Miyazaki et al., 2010b), which requires additional study to be confirmed.

Two questions arise from our results: **1)** what are the separate roles of the different planetary and synoptic-scale wave types within the total extratropical wave modulation of the TIL, and **2)** which is the mechanism for wave amplification near the tropopause as seen in Fig. 3 (see section 5).

Our study, working only with COSMIC GPS-RO observations, has identified and quantified an important mechanism for extratropical TIL enhancement: extratropical wave modulation, which is dominant in the extratropics and especially at mid-latitudes. In contrast with earlier literature that focused on TIL-enhancing mechanisms with permanent and irreversible effects, our wave modulation mechanism is reversible and transient, but constantly present. We suggest that the remaining TIL

in polar regions can be explained by accelerated residual circulation from polar vortex disturbances (given the similarities of our results with Wargan and Coy (2016)) and water vapor radiative effects in polar summer, although these would need to be confirmed by additional studies.

**Appendix A:  Wave modulation of the TIL in SH Mid-latitudes**

Figure A1 is the Southern Hemisphere equivalent of Fig. 4 (which was for 40°N). The signatures of the different extratropical waves in Fig. A1 lead to the same conclusions for the SH mid-latitudes: all defined waves show a net cold anomaly maximizing at the tropopause, a slight mean warm anomaly above it, and a net $N^2$ increase directly above the tropopause (Fig. A1 a-d). The strongest wave

signal belongs to eastward-propagating waves with periods of 4 to 25 days (red lines), which is even more dominant in Fig. A1 than in Fig. 4 due to the stronger westerlies found in the SH. Quasi-stationary waves (periods of 30-96 days, black and dashed magenta lines) and the s=0 wave (grey line) play a minor role in enhancing the TIL in both hemispheres.

The total extratropical wave signal (Fig. A1 e-f) at 40°S is a ~3.6K colder tropopause in the

750 seasonal zonal-mean, tropopause-based profile, and a TIL enhancement of ~$1.7 \times 10^{-4} s^{-2}$ in winter (red line). In summer (black line) the modulation is weaker: tropopause mean cold anomaly of ~3.0K and ~$1.4 \times 10^{-4} s^{-2}$ of TIL enhancement. The total extratropical wave signature in SH mid-latitudes has the same winter-summer seasonality as the NH, and a slightly higher magnitude throughout the year. The spring and autumn wave signatures are similar to the results shown here for winter and

755 summer, respectively.

Figure A2 compares the daily evolution of the zonal-mean, tropopause-based vertical $N^2$ profile at 40°S, with (Fig. A2a) and without the extratropical wave signal (Fig. A2b). Fig. A2 is the SH equivalent of Fig. 5, and also leads to the same conclusions: there is a distinct TIL in Fig. A2a throughout the year (~$6 \times 10^{-4} s^{-2}$, orange-red in winter; ~$5 \times 10^{-4} s^{-2}$ yellow in summer), which

is almost completely gone once the extratropical wave signal is subtracted. The weak hints of a TIL seen in Fig. 5b are even weaker in Fig. A2b, suggesting that other TIL enhancing processes play an even smaller role in the SH. We conclude from Fig. A2 that the TIL modulation by planetary and synoptic-scale waves explains most of the TIL strength in the tropopause-based $N^2$ structure at mid-latitudes also in the SH.

The findings of subsection 4.1 (NH mid-latitudes, Figs. 4 and 5) also apply to the SH (Figs. A1 and A2) in a nearly-coincident way.

*Acknowledgements.* This study was completed within the Helmholtz-University Young Investigators Group NATHAN project, funded by the Helmholtz Association through the president's Initiative and Networking Fund and the GEOMAR Helmholtz-Centre for Ocean Research in Kiel. We thank the ECMWF data server for

the freely available ERA-Interim data; and UCAR for the COSMIC, satellite missions' temperature profiles.

The assistance accessing different datasets and discussions with Sandro Lubis, Wuke Wang and Sebastian Wahl are also appreciated.

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

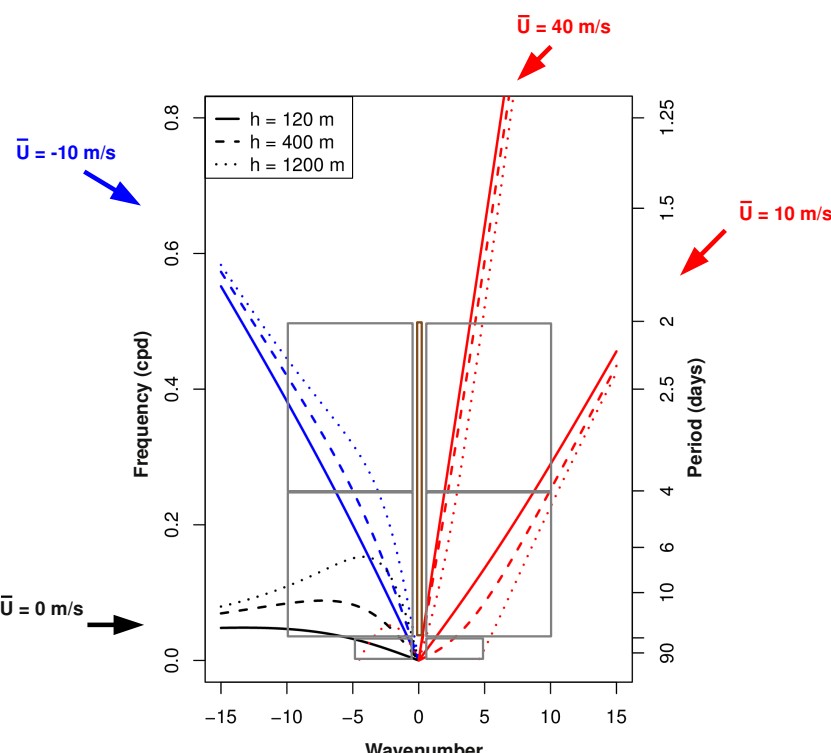

**Figure 1.** Dispersion curves for forced Planetary Waves at 50°N under different mean zonal wind regimes (line colors, winds specified outside the diagram), and differentiating equivalent depths (line type, top-left box). Filter bounds in the wavenumber-frequency domain are shown as grey boxes (brown for wavenumber zero).

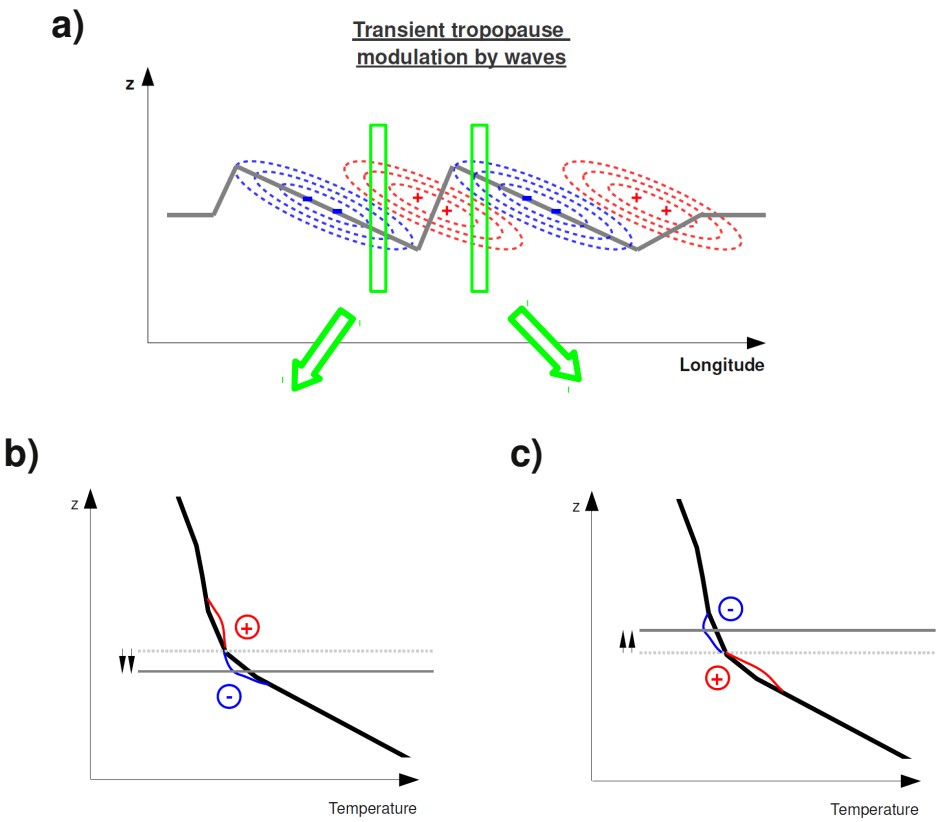

**Figure 2. a)** Schematic of transient tropopause modulation by an idealized wave with westward vertical tilt, as a snapshot of the wave's temperature anomalies (dashed contours: positive red, negative blue) and the undulating tropopause (thick and solid grey line). Green boxes represent the local samples for the bottom row. **b)** Ground-based mean temperature profile (thick black line, similar to the mid-latitude profile from Birner (2006), Fig. 8), zonal-mean tropopause height (grey dashed line), the local wave temperature anomalies superimposed on the zonal-mean temperature profile (positive red, negative blue), and the resulting locally lower tropopause (solid grey line, double arrow). **c)** same as (b) but with the opposite sign of the anomalies and a locally higher tropopause.

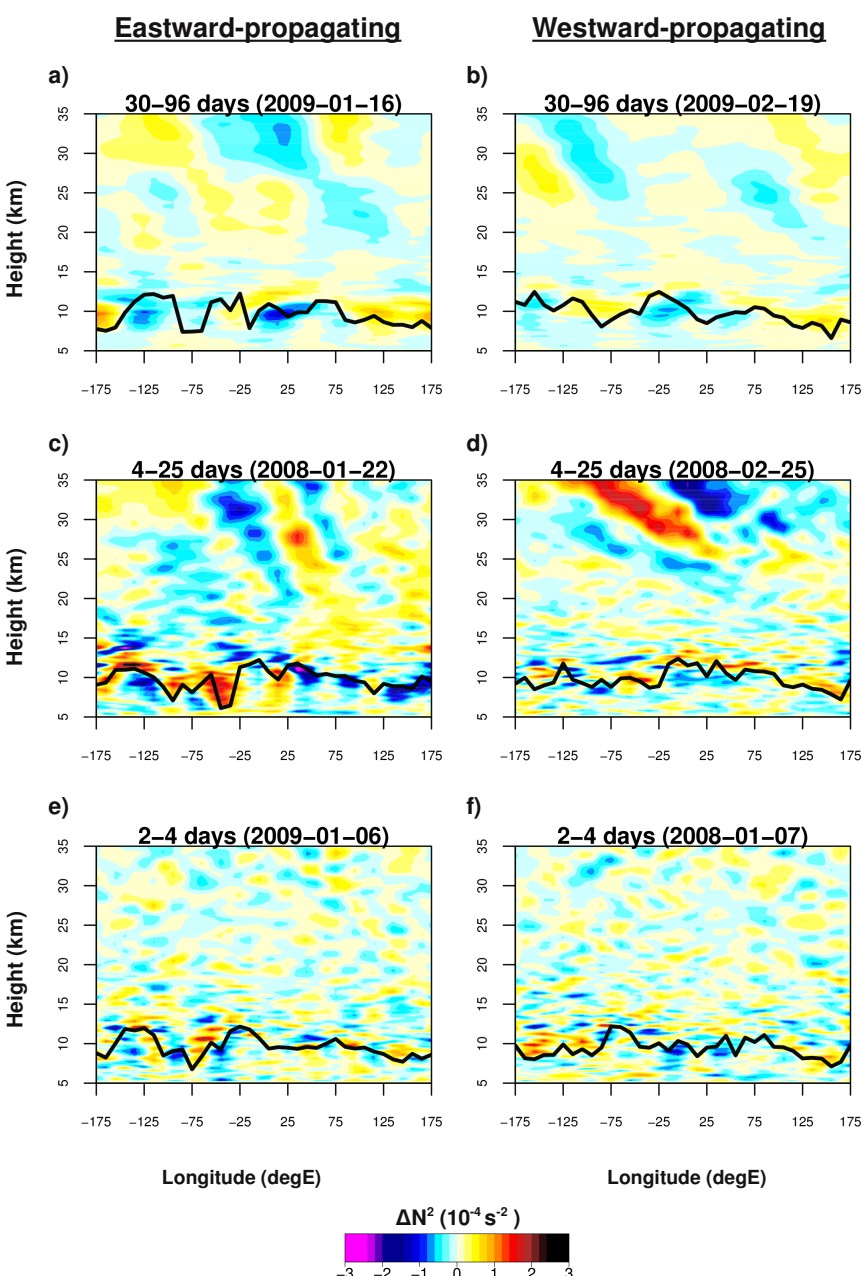

**Figure 3.** Longitude-height snapshots of the $N^2$ anomalies of the different wave spectral regions at certain dates, for the 50°N latitude band. The wave spectral regions correspond to the wavenumber-frequency domains defined in Fig. 1, except for wavenumber zero. Left column are eastward-propagating waves, right column are westward-propagating waves, and their periods are specified along with the date. The black line denotes the thermal tropopause from GPS-RO profiles.

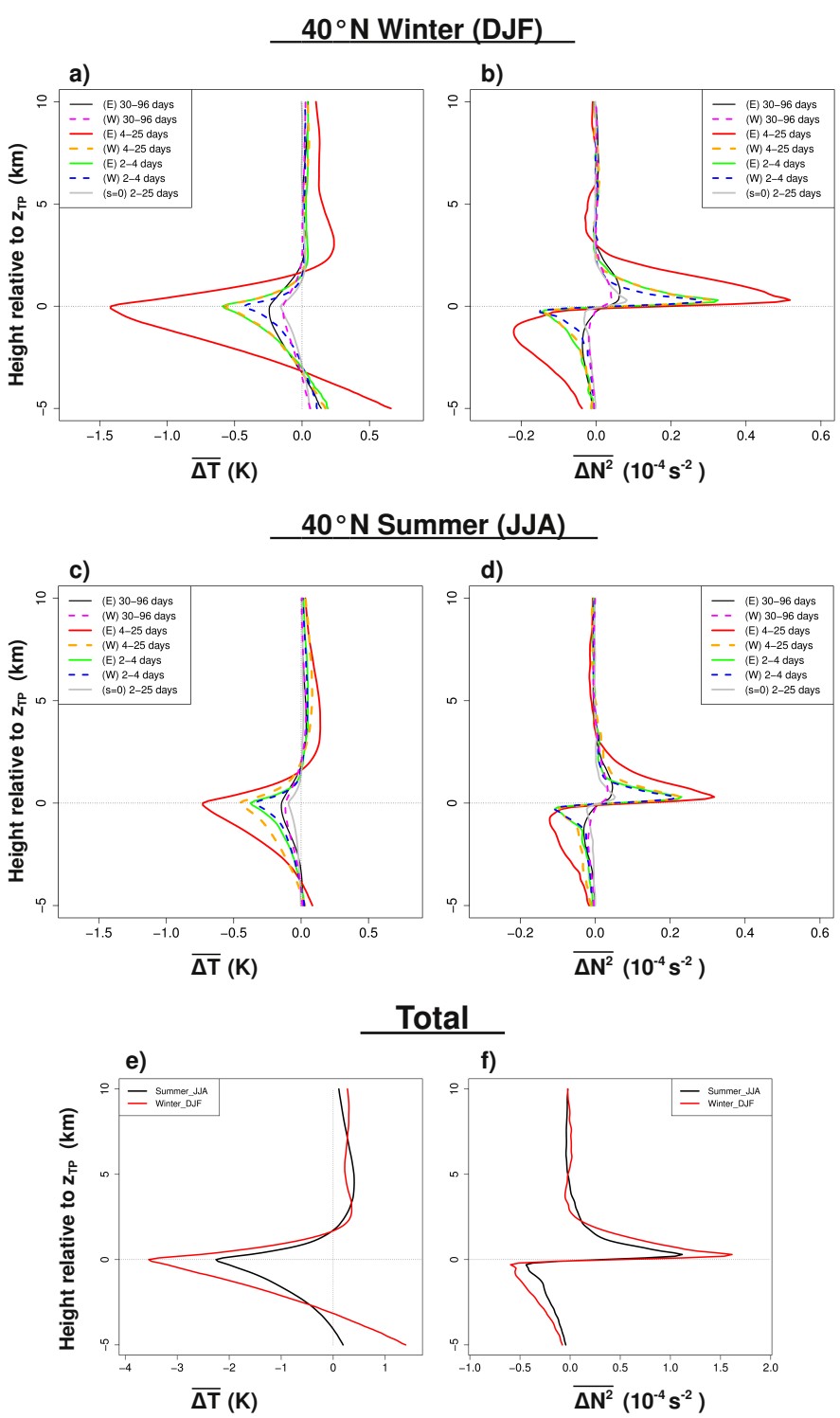

**Figure 4.** Average signature of the different wave spectral regions at 40°N, as the mean anomaly in the zonal-mean vertical profiles of temperature ($\overline{\Delta T}$, left column) and static stability ($\overline{\Delta N^2}$, right column). Top row (a and b) for winter (DJF), middle row (c and d) for summer (JJA). Bottom row (e and f) compares the total seasonal wave signatures.

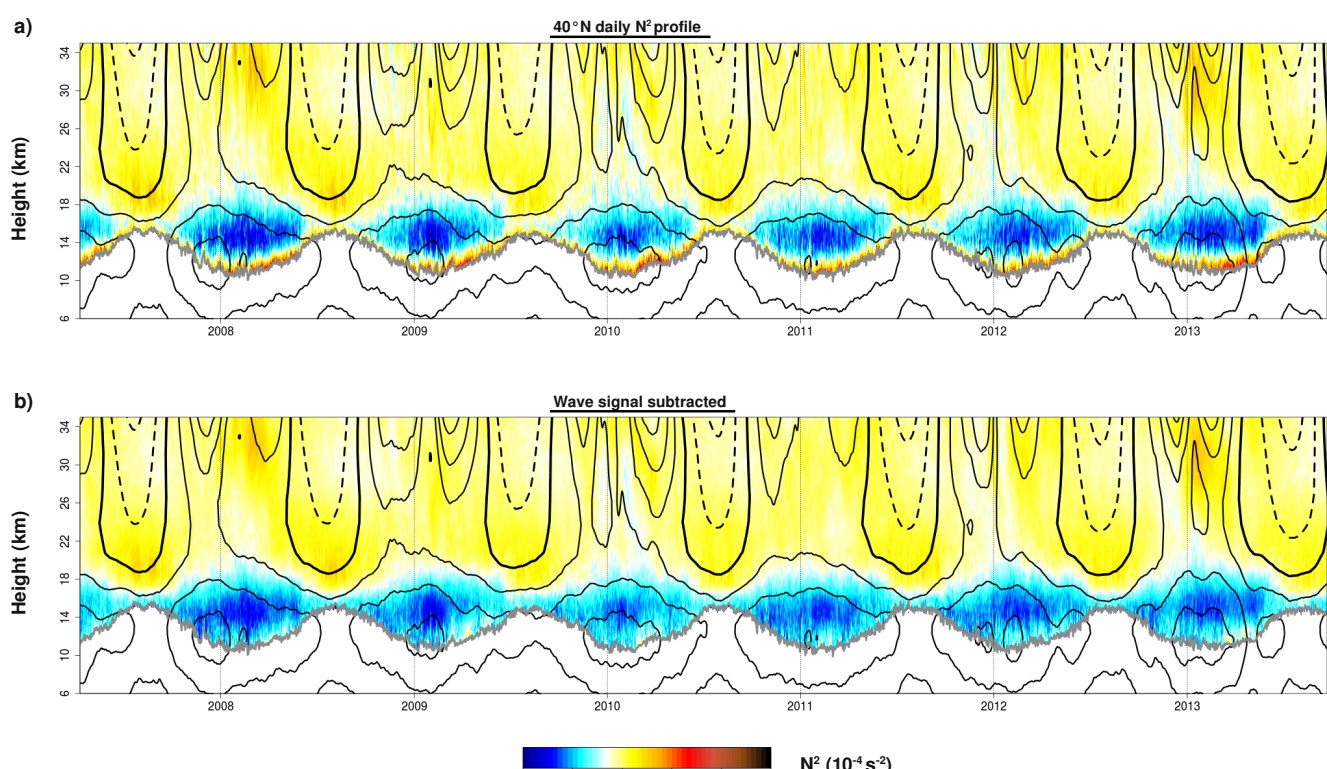

**Figure 5.** a) Daily evolution of the tropopause-based, 40°N zonal mean $N^2$ vertical profile between 2007-2013 (colors) from COSMIC GPS-RO profiles. The grey line denotes the tropopause height ($TP_z$). Thin black contours denote positive (westerly) mean zonal wind, with a thicker contour for the zero line, dashed contours for negative (easterly) winds, and a 10m/s separation. Winds were obtained from the ERA-Interim reanalysis. To improve visibility, the winds are displayed with a running mean of +-15 days. No running mean is applied to the $N^2$ vertical profile or $TP_z$ in order to allow the subtraction of the extratropical wave signal. Note that tropospheric $N^2$ values are not included in the color scale and therefore left blank. b) Same as in Fig. 5a, but the wave signal has been subtracted from the $N^2$ vertical profile.

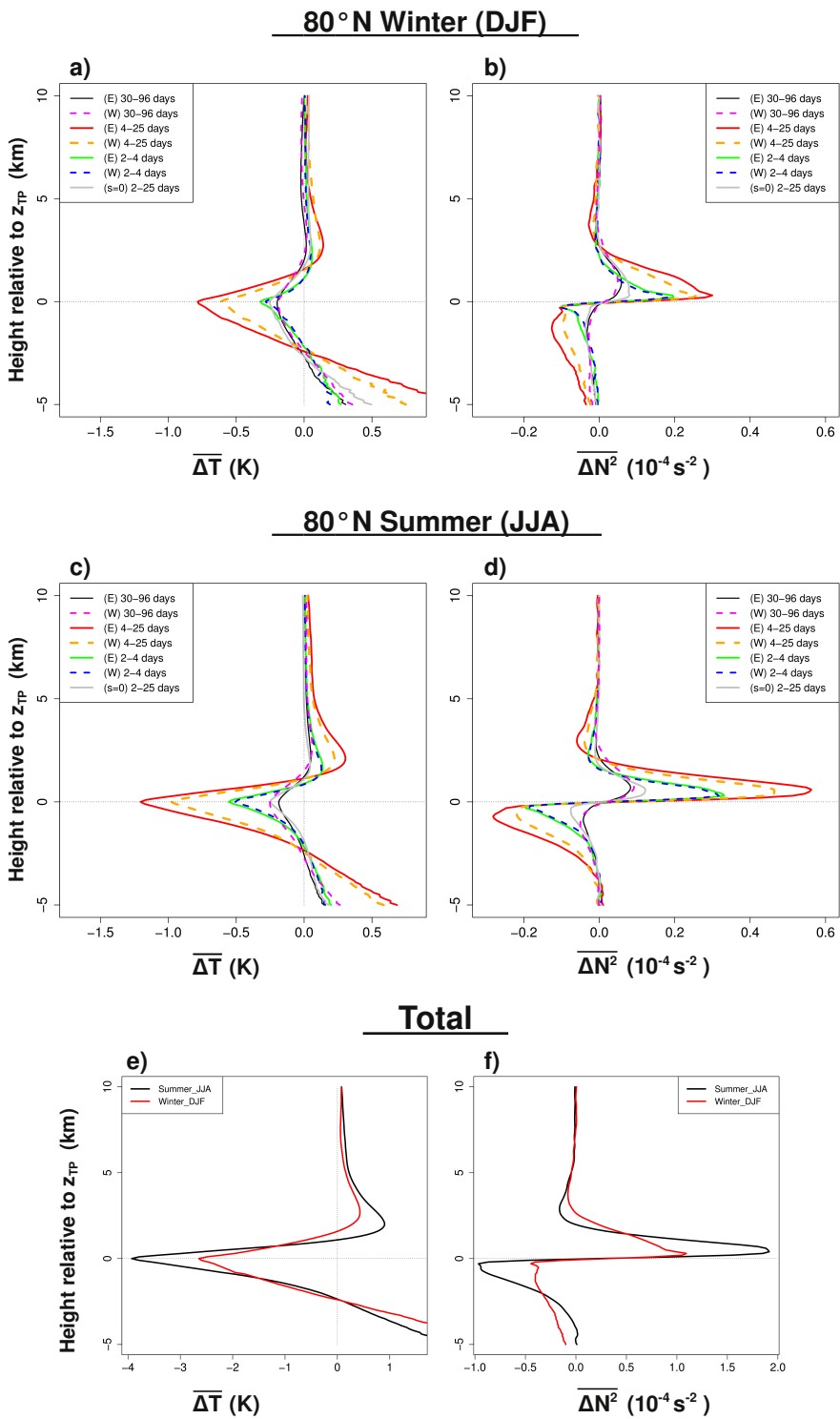

**Figure 6.** As in Fig. 4, but for 80°N.

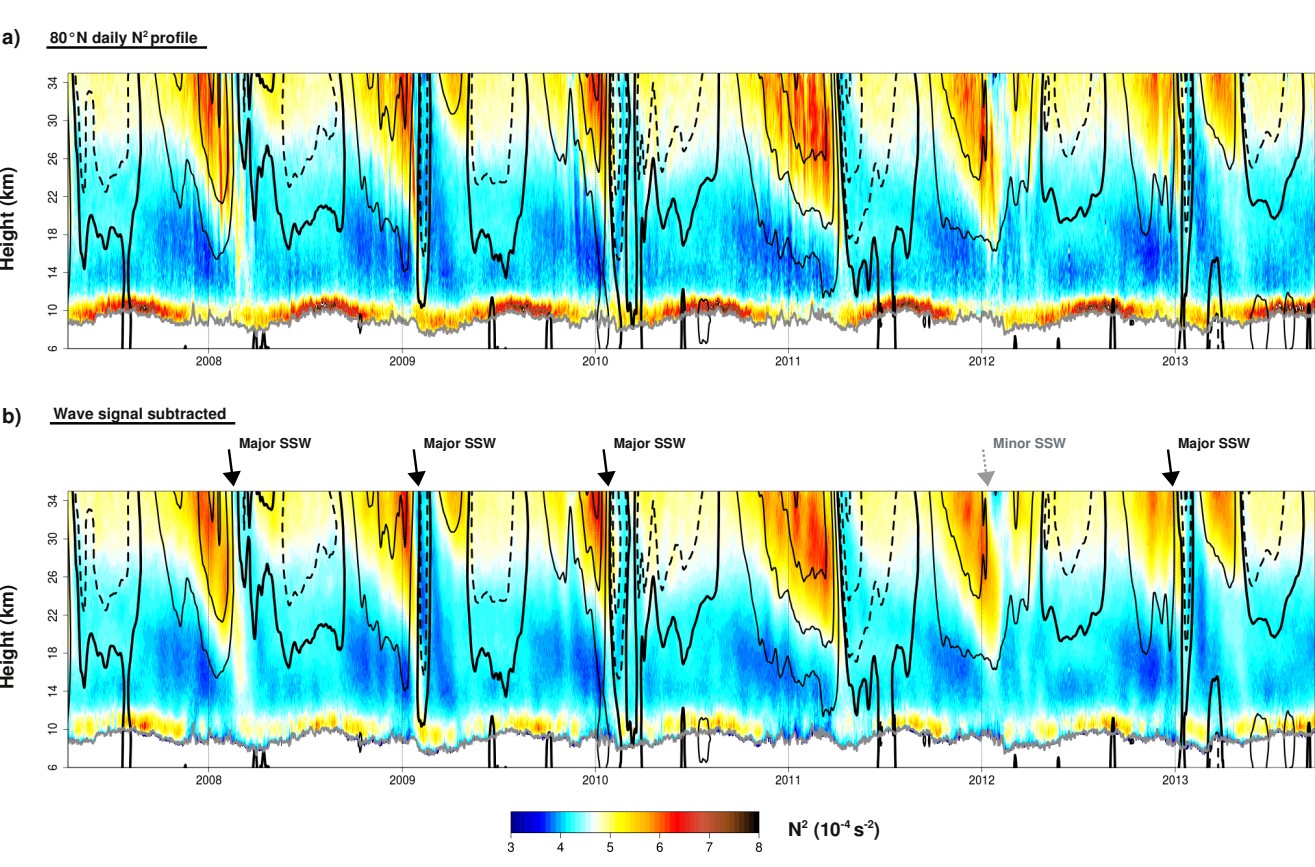

**Figure 7.** As in Fig. 5, but for 80°N. Major and minor SSWs are marked above Fig. 7b. First easterly wind contour (dashed line) at -3m/s for better visibility. The rest of wind contours are 10m/s intervals as in Fig. 5.

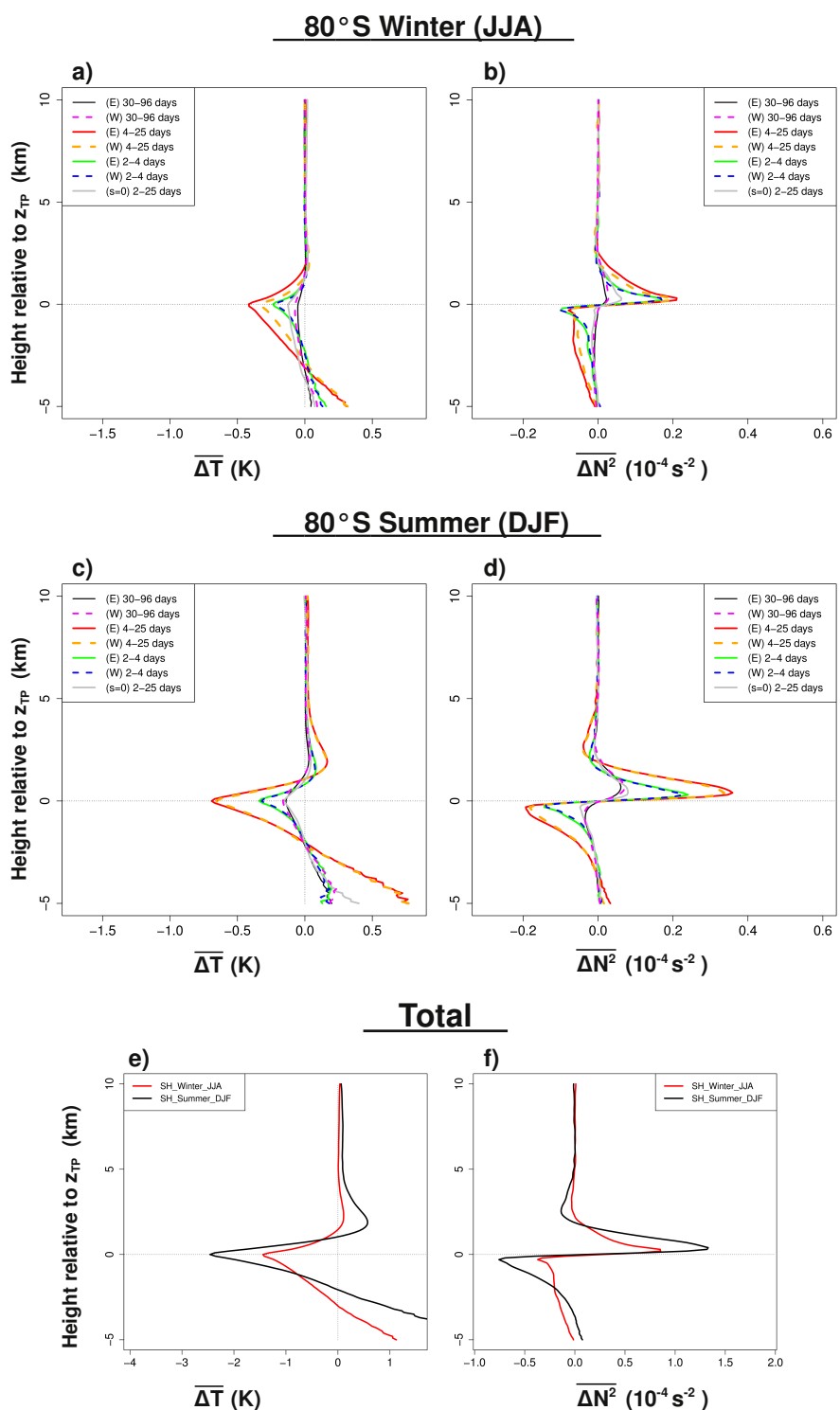

**Figure 8.** As in Figs. 4 and 6, but for 80°S.

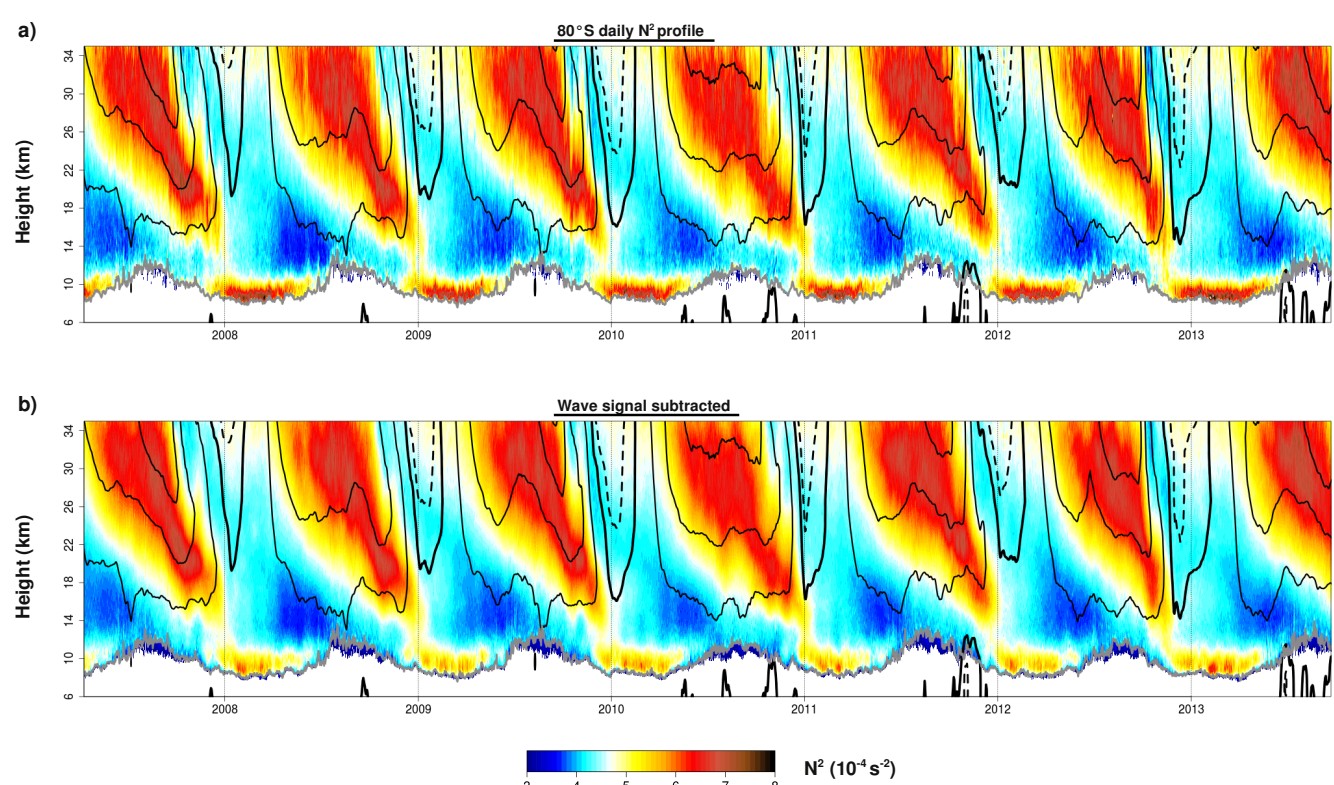

**Figure 9.** As in Fig. 5 and 7, but for 80°S. First easterly wind contour (dashed line) at -3m/s for better visibility. The rest of wind contours are 10m/s intervals as in Fig. 5.

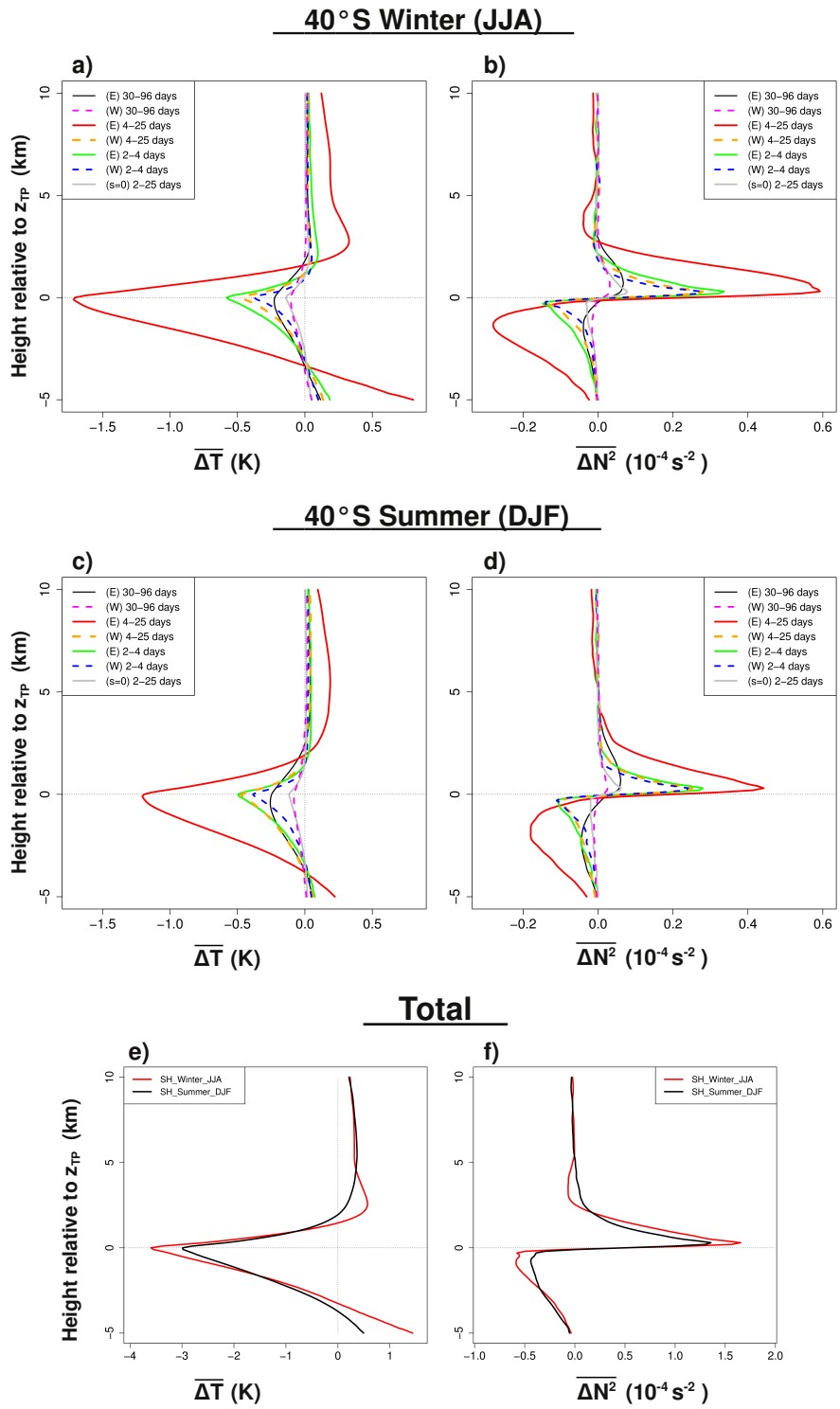

**Figure A1.** As in Figs. 4, 6 and 8, but for 40°S.

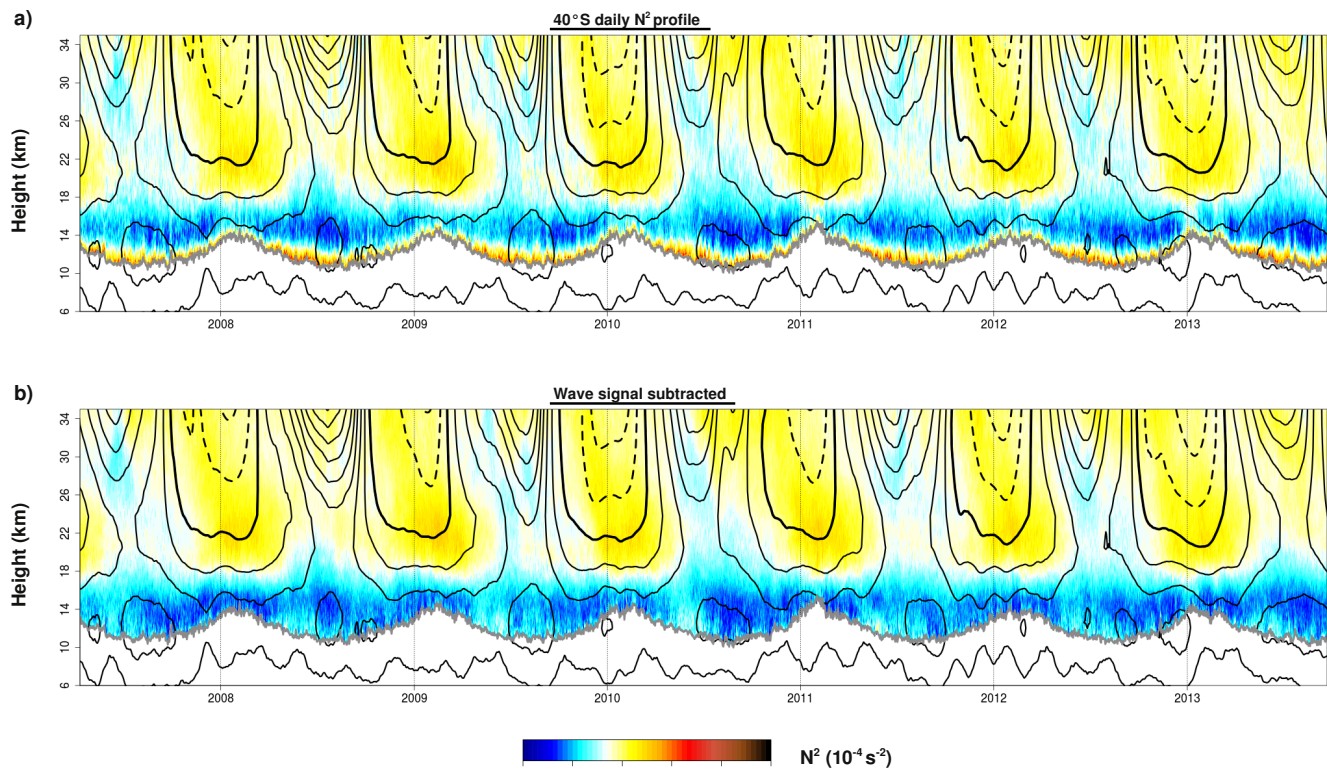

**Figure A2.** As in Fig. 5, but for 40°S.