# Peer review of "Wave Modulation of the Extratropical Tropopause Inversion Layer"

_Atmospheric Chemistry and Physics, 2016_

## Referee Comment (RC1) · Anonymous Referee #1 · 12 Oct 2016

General comments

Influences of planetary and synoptic-scale waves on strength of the tropopause inversion layer (TIL) were investigated in the mid/high-latitude regions using GPS radio occultation data. Two-dimensional wave filtering (in longitude and time) and tropopause-based average were made to isolate wave signal in temperature and static stability (N2). Authors showed that strength of the TIL (N2 right above the tropopause) decreases significantly after applying the wave filter, and attributed it to reduction of planetary and synoptic-scale wave signature in the extratropical TIL. They also presented influences of stratospheric sudden and finial warnings on N2 of the stratosphere (and near the tropopause). Research question is clear, and the manuscript is generally well written. However, the methodology used in this study has clear limitations in isolating midlatitude-wave signature in the TIL, and physical interpretations made in the results

are controversy. I have two major questions that authors may want to consider before publication of this manuscript.

Specific comments (major)

1. For filtering method and their physical interpretation:

Authors are using filtering limits of 2-4, 4-25, 30-96 days in time and -10 to 10 wavenumber in zonal direction. The choice of filter domain is subjective, and it is difficult to make a physical connection between the filter area and mid/high latitude waves. The filter seems to cover too broad spectral range, and it may remove most of the variability within 2-96 days. Particularly, the TIL could be easily removed by variation in tropopause height, because the TIL is defined in tropopause-relative coordinate and filtering is made in altitude coordinate. Therefore, a large portion of the "tropopause-based zonal-mean TIL (shown in Figs. 5, 7, and 9)" could be filtered out, NOT because it is a part of Rossby waves BUT because Rossby waves simply change height of the tropopause. This problem could be worse with a large filter area, and authors may want to test this issue.

I agree with authors that it is difficult to define general filter area (or filtering method) for mid-latitude waves. However, some efforts are still required to define reasonable filtering area in order to overcome the limitation and get more reliable results. Careful examination on wavenumber-frequency spectrums may be helpful for figuring out Rossy wave signature in the TIL, and some results of it will also be appreciated by readership.

Minor note:

Authors mentioned that they "define a seventh filter for wavenumber zero for completeness (page 7, line 229)". It sounds like "the filter used for Figs. 5, 7 and 9 also removes wavenumber zero along with wavenumber 1-10". If this is the case, wavenumber zero should not be filtered as it is a part of mean overturning circulation (i.e., deep or shallow

branch of the Brewer-Dobson circulation).

2. Structure of temperature anomaly:

The temperature structure shown in Fig. 2 (and discussed in several parts in the manuscript) is more similar to that of gravity waves rather than that of Rossby waves. Although vertically propagating Rossby waves (with wavenumber 1, 2) have westward tilt, the tilting is not big enough to make strong N2 modulation at the tropopause (see Fig.7 of Fletcher and Kushner 2011 for example). Synoptic-scale waves could make strong modulation in temperature and N2 near the tropopause, but their temperature structure is different from that shown in Fig. 2. Based on Hoskins et al. (1985, their Fig. 15), the (potential) temperature and inferred N2 anomalies have horizontally flat structure near the tropopause. It show dense packing of isentropes over high pressure system implying strengthening of the TIL over anticyclones (this feature is constant with the idea of vertical convergence). If authors think that transient waves in 4-25-day frequency band are major contributor and Fig. 2 is their common temperature structure, it should be obtained from observation (not from conceptual figure).

I do not disagree with authors' main idea that planetary and synoptic-scale waves could have a major contribution in making strong extratropical TIL. However, I recommend that authors should test and interpret their results more carefully because the results can be sensitive to the methodology used for filtering of wave signature.

I still think that this is well-organized study and documentation. I believe authors can overcome (or minimize, at least) the limitation through careful examination of the wave spectra.

References

Fletcher, C. G., & Kushner, P. J. (2011). The Role of Linear Interference in the Annular Mode Response to Tropical SST Forcing. Journal of Climate, 24, 778–794. http://doi.org/10.1175/2010JCLI3735.1

Hoskins, B. J., McIntyre, M. E., & Robertson, a. W. (1985). On the use and significance of isentropic potential vorticity maps. Quarterly Journal of the Royal Meteorological Society, 111(470), 877–946. http://doi.org/10.1002/qj.49711147002

Interactive
comment

---

## Referee Comment (RC2) · Anonymous Referee #2 · 19 Oct 2016

ACPD REVIEW

TITLE: Wave Modulation of the Extratropical Tropopause Inversion Layer AUTHORS: Robin Pilch Kedzierski, Katja Matthes, and Karl Bumke

Summary: This paper examines the influence of waves on the middle latitude and polar tropopause, particularly the increase in static stability often found just above the tropopause, (TIL, tropopause inversion layer). The analysis of tropopause structure is based on a detailed analyses of GPS RO observations. Results show that most of the extratropical TIL variability can be explained by synoptic scale Rossby waves. Results also show how changes in stability propagate downward during SSW events. However, the authors also note that there is still significant unexplained variability, especially in the NH polar region. The paper is well organized. The abstract state both the problem

and the main results. The introduction references and reviews appropriate material. The methodology is explained in detail. Results are clearly presented by hemisphere and latitude. Overall the is an excellent paper that should be of interest to many readers of ACP.

Overall Recommendation: Publish after minor changes.

Major Comments:

Line 278: Regarding Fig. 3: Are the tropopause zonal structures calculated directly from the GPS-RO or from the tropopause-base coordinate? Does it make any difference? Also, if these are zonally averaged tropopause heights, it should state so in the Fig. 3 caption.

The paper mentions summer/winter differences in discussing Figs. 5, 7, and 9, yet it appears as if some the largest TIL events occur more in the spring, assuming the year tick marks denote the start of the new year. Would it be useful to discuss the results in terms of four seasons instead of just two?

Minor Comments:

Line 2: Probably should be something like "…how much amplitude and variability of the Tropopause Inversion Layer (TIL) comes from modulation by…" (optional)

Line 10: Once again: "The instantaneous modulation…" of the tropopause?, the TIL? I think it would read better if it explicitly stats what is being modulated. (optional)

Line 12: "…minor importance for the TIL amplitude and variability…" (optional)

Line 29: "…all latitudes…" The paragraph begins with discussing the extratropical TIL so does "all latitudes" include the tropics too or just extratropics?

Line 120: The paragraph beginning on line 120 starts with ERA-Interim and ends discussing GPS-RO. It seems awkward. Also, this appears to be the only mention of the ERA-Interim winds. It might improve the reading to tell how the ERA-Interim winds will

be used to place the GPS-RO observations into a broader context by identifying SSW events and seasonal wind changes. It also might be a good idea to acknowledge the source of the winds explicitly in the appropriate figure captions as well. (optional)

Line 165: "...all..." Weather sometimes changes more rapidly than 2-days. Maybe the "all" should be qualified to "most of" or "the majority of". (optional)

Line 194: "...and frequency..." "and lower frequency" maybe

Line 333: "...highest period..." maybe "longest period" would read better.

Line 368: "...TIL is clearly discernible in winter..." Maybe eliminate the word "clearly", as these features are relatively small in Fig. 5. Also, the TIL may be more distinct in spring than in winter (see second major comment).

Line 529: "...clear TIL during summer months..." Once again, maybe eliminate the work "clear", and the timing of the maximum TIL may be more toward spring than summer, but it is difficult to tell from the figure.

Other:

Some acronyms/variables are not defined where first introduced. These include: COSMIC GPS-RO (line 5), z (line 115), TPz (line 117), WMO (line 17), NCL (line 173).

---

## Referee Comment (RC3) · Anonymous Referee #3 · 20 Oct 2016

**Review of the manuscript** "Wave Modulation of the Extratropical Tropopause Inversion Layer" by R. P. Kedzierski, K. Matthes and K. Bumke, submitted for publication to ACP

**Recommendation:** Significant revisions

This paper investigates the impact of synoptic- and planetary-scale waves on the static stability structure of the extratropical tropopause region. This is done by using high-resolution temperature profiles from satellite measurements and applying wavenumber-frequency filtering to them. The technique is similar to the method applied previously by the authors to the tropical tropopause layer. In the present analysis, the extratropical Tropopause Inversion Layer (TIL) appears as largely due to these waves. In polar latitudes, the part of the TIL which cannot be associated with the waves is interpreted as due to either the residual circulation or to radiative effects. These results seem to be consistent with results from previous work.

The paper is generally well written. I have a few major and a number of minor comments which the authors should account for in revising the manuscript. In particular regarding their main premise of how waves affect the TIL, I have a few questions which should be clarified in order to make the argument consistent.

**Major comments**

1. A key aspect of this whole paper is the recognition that the purely adiabatic impact of a wave on static stability in the tropopause region and the concomitant modification of the tropopause height may lead to a systematic TIL enhancement. Figure 2 serves as a motivation and explanation. However, I did not quite understand the related description in lines 258–260, which seems to present a key argument in this context. Please clarify.

   Several times in the text (e.g. bottom of page 9) the authors stress that they typically expect a westward phase tilt with height corresponding to upward propagation of waves. This may be OK for planetary waves, but synoptic-scale waves tend to be evanescent in the stratosphere, so I would not expect much of a phase tilt in the lower stratosphere. In addition, wave modes which represent baroclinic instability (like in the Eady model) do have a westward tilt with altitude regarding perturbation geopotential or the perturbation streamfunction, but the associated tilt of the temperature anomaly actually has an *eastward* tilt with height (see Fig. 8.10 in the textbook of Holton 2004). Also, it has been argued in the past that some gravity wave activity is actually generated in and, hence, emanates from the tropopause region (O'Sullivan and Dunkerton, 1995), and the phase tilt of such waves seems less clear or may be insignificant. How does this affect the arguments presented in the paper, which evoke a westward tilt with altitude?

2. Fig. 3 shows a few selected cases. Are these random picks, or are these cases where the wave signal at the tropopause level can be seen best. If the latter is true, this would mean that often the wave signal may be rather incoherent and hard to interpret?!

3. In their interpretation sections, the authors could tie their results more comprehensively to earlier studies, including model studies, in particular the one by Miyazaki *et al.* (2010), who systematically quantified different mechanisms for TIL formation in a GCM including their hemispheric and seasonal behavior. Also, to what extent does the currently analysis (implicitly or explicitly) include the traveling cyclones and anticyclones which have been argued before to play a role for TIL formation, and to what extent are the current results consistent with the mechanisms suggested earlier? authors choose to leave this as a remaining question (see Page 12), but it could at least be formulated explicitly as a question.

4. I do not like the use of the terms "warming" and "cooling". The authors consider band-with filtered wave signals, in other words anomalies from the zonal mean. If a plot shows a local warm or cold anomaly, this does not necessarily imply that there is/was warming or cooling. It could just as well be the result of horizontal advection, i.e. the original air was replaced by warmer or colder air, but this does not imply any warming or cooling (neither diabatic nor adiabatic). Using more precise terminology would make the discussion of the processes more lucid.

**Minor issues**

1. Line 86: replace "high amounts of ...." by a more idiomatic expression.

2. Line 110: the 100 m vertical resolution of GPS-RO temperature profiles, is this really comparable to the vertical resolution of radiosondes? I thought that the latter have even significantly higher resolution. To be sure, I believe that 100 m resolution is sufficient for the purpose of the current paper.

3. Line 126: replace "grid with 10° separation" by a better expression (grid points may have a separation, but not the grid itself).

4. Line 128: replace "exponentially-folding function" by "Gaussian function"

5. Line 153: replace "+- 1 longitude grids" by a better expression (and same with time-discretization)

6. Line 157: replace "in the equator" by "on/at the equator"

7. Line 160: replace "lower bottom of the vertical scale" by "lower bound of the vertical scale"

8. Line 180: replace "remaining of this section" by "remainder of this section"

9. Line 203: this dispersion relation seems to be the version for the shallow water system, right? If so, this should explicitly be stated, and it should also be motivated to what extent this can be used in the current context. After all, the paper deals with the vertically stratified atmosphere, *not* with the shallow water model. The term "equivalent depth" should then be explained, and also in what sense the term $f^2/gh$ can be considered as an "approximation to account for vertial propagation".

10. Bottom of page 7: I find it misleading to talk here about "different wave types". As the authors show quite clearly, there is a broad spectrum of waves, which *cannot* be classified into different *types* as in the tropics. This is, by the way, exactly what the authors say themselves on page 10 ("unable to differentiate particular wave types"). Maybe one should simply refer to "different parts of the spectrum".

11. Line 236: replace "is outstanding" by a more idiomatic expression.

12. Line 249: replace "associated to" by "associated with".

13. Line 267: replace "would remain the same despite the presence of wave anomalies" by "are unaffected by the wave anomalies"

14. The longitude-height sections of Fig. 3 could be improved by actually ploting all the way from -180 ° to +180 ° .

15. Line 360: In what sense it $s = 0$ a wave?

16. Line 511: I would try to avoid the abbreviation "mSSW".

17. Line 611: replace "humbler" by "smaller"

18. Figure 2: put axes labels, i.e. longitude (or $\lambda$) and altitude (or $z$).

19. In figure 5 (and all corresponding figures showig time versus altitude) one might consider ploting only up to 30 km, without changing the vertical extent of the plot. This would give somewhat better resolution in the vertical, putting more emphasis on the features around the tropopause region, which is relevant here. On the other hand, the information way up in the stratosphere is not very relevant for this paper (except where discussing stratospheric warmings).

**References**

Holton, J. R. 2004. *An Introduction to Dynamical Meteorology*. Elsevier Academic Press, 529 pp., fourth edition.

Miyazaki, K., S. Watanabe, Y. Kawatani, Y. Tomikawa, M. Takahashi, and K. Sato 2010. Transport and mixing in the extratropical tropopause region in a high-vertical-resolution GCM: Part I: Potential vorticity and heat budget analysis. *J. Atmos. Sci.* **63**, 1293–1314.

O'Sullivan, D., and T. J. Dunkerton 1995. Generation of inertia-gravity waves in a simulated life cycle of baroclinic instability. *J. Atmos. Sci.* **52**, 3695–3716.

---

## Short Comment (SC1) · 1 Nov 2016

General comments

The article under discussion provides a compelling demonstration of the role of synoptic and planetary wave activity in the (strength of the) extratropical TIL. However, the article's abstract and conclusions leave the overall impression that the extratropical TIL is unrelated to radiation. I think the authors would agree that their conclusion that the zonal-mean extratropical TIL is modulated over time by large-scale waves does not exclude the radiative forcing of the TIL; it just identifies the main wave-dynamical effect on the zonal-mean state of the TIL.

(In the following, literature citations refer to the paper reference list, with the exception of the two references given below.)

[Figure]

FDH radiative calculations showed that the extratropical TIL, on average, depends strongly on radiative cooling of the tropopause by water vapor, or on the distribution of water vapor near the tropopause, in midlatitudes (Randel et al., 2007; Kunz et al., 2009) and in polar latitudes (Randel and Wu, 2010).

Dry synoptic-scale dynamics in adiabatic-inviscid flow models accounts for the formation of the extratropical TIL, without simulating its actual strength (Wirth, 2003; Wirth and Szabo, 2007; Son and Polvani, 2007; Erler and Wirth, 2011). Moreover, as the authors probably know, the role of moisture and radiation (mostly the vertical gradient of specific humidity) was eventually recognized to be important in synoptic-scale modeling of the TIL strength; see ref. [1]

Somewhat in between the dynamical and radiative views, another study showed that both thermal radiation and large-scale dynamics are important to the seasonal-mean midlatitude TIL; see ref. [2]. Aiming to understand the persistency of the extratropical TIL, that study not only explained how the vertical gradients of water vapor around the tropopause are key for the average seasonal TIL, but also showed that the midlatitude TIL strength depends not less on local dynamical warming.

Considering the increasing evidence from previous studies, in my opinion, the present article would benefit from adding as a caveat that radiation is likely to play its own role in the extratropical TIL, even if coupled (by some process to be determined by future research) with the synoptic- and planetary-scale waves that modulate the zonal-mean TIL; the problem is that the concluding section only remarks the radiative cooling of the polar summer tropopause. Small improvements concerning the Abstract are suggested below in the specific comments.

Additionally, the literature citations, at least in the introductory section, might be up to date.

[1] Kunkel, D., Hoor, P., and Wirth, V.: The tropopause inversion layer in baroclinic life-cycle experiments: the role of diabatic processes, Atmos. Chem. Phys., 16, 541-560,

doi:10.5194/acp-16-541-2016, 2016. // First published in Jan 2016

[2] Ferreira, A. P., Castanheira, J. M. and Gimeno, L. (2016), Water vapour stratification and dynamical warming behind the sharpness of the Earth's midlatitude tropopause. Q.J.R. Meteorol. Soc., 142: 957–970. doi:10.1002/qj.2697. // First published in Dec 2015

Specific comments

Line 4: the phrase "it also puts other TIL enhancing mechanisms into context" should read as "it also puts other TIL enhancing dynamical mechanisms into context", since radiative effects are not addressed in the study.

Lines 10-11: instead of "The instantaneous modulation by planetary and synoptic-scale waves is almost entirely responsible for the TIL in mid-latitudes", it would be more accurate to say: "Planetary and synoptic-scale waves are almost entirely responsible for the instantaneous modulation of the TIL in mid-latitudes". In this way, the role of radiation is not left out, while keeping the authors' main finding (wave modulation of the mid-latitude TIL).

Lines 18-20: "After many modelling studies (...) in the last decade, our study finally identifies which processes dominate the extratropical TIL strength and their relative contribution, by analyzing observations only." The sense of this sentence is questionable because the present and previous works on the extratropical TIL have dealt with different time and/or spatial scales, so they are not easily comparable; besides the FDH sensitivity tests supporting the radiative hypothesis are a mix of observations and radiative modelling. I am sure that the authors do not want to say that the previous theories are all marginal to the subject. Then, it would be more constructive to write something like, "In addition to the TIL enhancing mechanisms proposed by modeling studies in the last decade, our study now identifies which dynamical processes dominate the zonal-mean extratropical TIL strength and their relative contribution, by analyzing observations only."

Lines 363-364: it would be helpful to the reader to clarify the expression "TIL enhancement" (from the values given in the text, it refers to the increase of $N^2\_max$). Since it is used more than once, it should be explained in the introductory part of section 4.

Lines 380-381: when saying that the TIL is "almost completely gone" after subtracting the extratropical wave signal, the authors meant to say that $N^2\_max$ is greatly reduced. This might be clarified in parenthesis.

Lines 501-502: "The only other mechanism restricted to polar summer that could enhance the TIL is water vapor radiative cooling of the tropopause". The word "restrictive" seems misleading here. I would suggest rephrasing in this way: "The only other mechanism that could enhance the polar summer TIL is water vapor radiative cooling of the tropopause".

Lines 559-560: Since the text refers specifically to "the remaining TIL", I would suggest to replace "is enhanced by" –> "is due to"

Figures 5, 7, 9: in Figure 5, we expected to see $N^2$ values in the range 1-2 $\times 10^{-4}$ $s^{-2}$ in the upper troposphere. So, I don't understand why $N^2$ is shown in white (or is not shown) below the tropopause. The same applies to Figures 7 and 9, even if the high-latitude upper-troposphere static-stability values are different. I guess this was done to enhance the color map in the lower stratosphere. Anyhow, values not shown in the plots should be mentioned at some point (e.g., figure caption of Figure 5).
* * *

---

## Author Comment (AC1) · 14 Dec 2016

**Response to reviews of Pilch Kedzierski et al.: "Wave Modulation of the Extratropical Tropopause Inversion Layer"**

Dear Editor,

We would like to thank the three reviewers and A. P. Ferreira for their helpful comments. The manuscript has benefited greatly from their suggestions, especially in the more detailed description of our filtering method, the wave modulation mechanism, and the discussion section linking it to previous literature. The manuscript has gained length significantly, but we feel that it has also gained in clarity and integration within TIL literature.

In the following paragraphs we include our point-by-point response to each comment in the reviews along with the changes made in the manuscript. The referee's comments are in blue font, and our replies are in normal font. Every change made in the revised manuscript is highlighted. We hope that the new version fulfills the reviewer's requests.

Yours sincerely,

Robin Pilch Kedzierski
Katja Matthes
Karl Bumke

**General comments**

Taking into account all comments, we realized that most of them are grouped around two main issues:

1) The filtering method and the wave modulation mechanism. These needed more detailed explanations, which have been expanded. Especially regarding our wave modulation mechanism, we added two panels to Figure 2 (b and c now), and we separated the explanation of the mechanism into a new subsection 2.4 (it was part of subsection "2.3: Wavenumber-frequency domain filtering" before).

2) The interpretation of our results and their relation to earlier studies and previously proposed mechanisms. In the submitted manuscript we did not discuss this, and we agree with the comments that our paper would benefit greatly from such discussion. Now, we created a new discussion section 5, linking the wave modulation mechanism to isentrope packing above anticyclones (Wirth 2003, 2004), wave breaking (Erler and Wirth 2011) and radiative effects from water vapor / clouds (Miyazaki 2010, Kunkel et al. 2016, Ferreira et al. 2016). In this

detailed discussion we suggest that all these mechanisms, directly or indirectly, can integrate into the amplified extratropical wave whose effect is adiabatic and transient. In our opinion, the important benefit of our method is that it separates this transient effect, as defined with space-time harmonics of T and N2 anomalies whose sum is zero in the ground-based mean, from the rest. The diabatic and irreversible processes from radiation and residual circulation, and their direct influence on the TIL, are still present in Figs. 7b and 9b once our wave anomalies are subtracted.

We stress that ultimately the direct effect on the TIL is done by extratropical waves (see the new section 5 for the detailed discussion) in an adiabatic and transient way, and the remaining question is what causes the waves to gain large amplitudes. We suggest that some process, possibly a radiative-dynamical feedback, is helping wave growth near the tropopause

**Point-by-point responses to Ref#1**

Specific comments (major)
1. For filtering method and their physical interpretation:
Authors are using filtering limits of 2-4, 4-25, 30-96 days in time and -10 to 10 wavenumber in zonal direction. The choice of filter domain is subjective, and it is difficult to make a physical connection between the filter area and mid/high latitude waves. The filter seems to cover too broad spectral range, and it may remove most of the variability within 2-96 days. Particularly, the TIL could be easily removed by variation in tropopause height, because the TIL is defined in tropopause-relative coordinate and filtering is made in altitude coordinate. Therefore, a large portion of the "tropopause-based zonal-mean TIL (shown in Figs. 5, 7, and 9)" could be filtered out, NOT because it is a part of Rossby waves BUT because Rossby waves simply change height of the tropopause. This problem could be worse with a large filter area, and authors may want to test this issue.

We added several additional paragraphs throughout subsections 2.3 and the new 2.4 in order to clarify the filtering method, why it is optimal for the purpose of this study, and how the wave modulation mechanism works. No filtering is done in the vertical dimension, each vertical level is filtered separately (the filters work with longitude-time arrays, see subsection 2.2).

Rossby waves (and other types) modulate the T and N2 structures around the tropopause, changing the tropopause height and enhancing the TIL: there is no chain of causality here, it all happens simultaneously and only due to the presence of the wave. Apart from the increased detail in subsections 2.3 and 2.4, we further discuss the interpretation of our results in the new section 5.

We stress that the choosing of a broad spectral range is in order to capture as much wave variability as possible. If waves are not present, they won't have any effect on the tropopause-based means (l. 259-264), so a broader and partly empty spectral range is not a disadvantage.

Also note that, after subtracting the wave signal (see Figs. 7b and 9b), the remaining TIL and its timing with processes that are more permanent and irreversible (accelerated residual circulation during SSWs, and radiative effects from water vapor in polar summer) is in very good agreement with previous TIL literature.

I agree with authors that it is difficult to define general filter area (or filtering method) for mid-latitude waves. However, some efforts are still required to define reasonable filtering area in order to overcome the limitation and get more reliable results. Careful examination on wavenumber-frequency spectrums may be helpful for figuring out Rossy wave signature in the TIL, and some results of it will also be appreciated by readership.

We now added several paragraphs throughout subsections 2.3 and 2.4 (new) explaining more in detail why our filtering method is optimal for the purpose of the study. Regarding the detailed examination of the extratropical wave spectrum: this implies a similar study as Wheeler and Kiladis (1999), an update for the extratropics. Such study would itself yield a separate and abundant paper, and is well beyond the scope of the present manuscript that focuses on the TIL and its formation mechanisms; not on particular properties of atmospheric waves.

Minor note:
Authors mentioned that they "define a seventh filter for wavenumber zero for completeness (page 7, line 229)". It sounds like "the filter used for Figs. 5, 7 and 9 also removes wavenumber zero along with wavenumber 1-10". If this is the case, wavenumber zero should not be filtered as it is a part of mean overturning circulation (i.e., deep or shallow branch of the Brewer-Dobson circulation).

The s=0 filter represents zonally-symmetric, annular oscillations in time (any possible pole-trapped wave mode). We include this now in line 245-246 . Note that for a wave to be captured by this wavenumber-frequency domain filter, it needs by definition to oscillate regularly in the time dimension, which the Brewer-Dobson circulation does not. Also note that N2 anomalies associated with accelerations of the BD circulation (during polar vortex disturbances) in Fig. 7b are clearly present even after subtracting all the wave anomalies.

2. Structure of temperature anomaly:
The temperature structure shown in Fig. 2 (and discussed in several parts in the manuscript) is more similar to that of gravity waves rather than that of Rossby waves. Although vertically propagating Rossby waves (with wavenumber 1, 2) have westward tilt, the tilting is not big enough to make strong N2 modulation at the tropopause (see Fig.7 of Fletcher and Kushner 2011 for example). Synoptic-scale waves could make strong modulation in temperature and N2 near the tropopause, but their temperature structure is different from that shown in Fig. 2. Based on Hoskins et al. (1985, their Fig. 15), the (potential) temperature and inferred N2 anomalies have horizontally flat structure near the tropopause. It show dense packing of isentropes over high pressure system implying strengthening of the TIL over anticyclones (this feature is constant with the idea of vertical convergence). If authors think that transient waves in 4-25-day frequency band are major contributor and Fig. 2 is their common temperature structure, it should be obtained from observation (not from conceptual figure).

In the new subsection 2.4, we explain the wave modulation mechanism in more detail now. This mechanism applies to waves with any tilt. As long as they are large enough, even wave anomalies with vertical or flat tilt can modulate the tropopause height within a certain longitude range, and therefore have an effect on the tropopause-based seasonal zonal mean profile. Both our filtering method and the wave modulation mechanism are valid for any vertical tilt of the waves. We now see that this wasn't highlighted enough in the previous manuscript.

In the conceptual Fig. 2 we sketch an idealized case to show clearly how the wave modulation mechanism works with the most common extratropical wave type, the upward-propagating Rossby wave. Fig. 3 then shows how this looks like from GPS-RO observations, and particularly Fig. 3c is very similar to the sketch from Fig. 2a which is highlighted in the text. The wavenumber-frequency domain band shown in Fig. 3c is then the dominant one in the seasonal, tropopause-based mean signature of the waves in Figs 4, 6, 8 and A1, which we again link to Rossby waves several times in section 4.

In the new section 5 (Discussion) we link our wave modulation mechanism to earlier literature about cyclone-anticyclone and isentrope packing, among other previously proposed mechanisms.

Regarding the references to Hoskins et al. (1985, their Fig. 15) and the experiments by Fletcher and Kushner (2011):

Axisymmetric vortices are found seldom in the real atmosphere, the closest examples being tropical cyclones or the centers of very strong extratropical cyclones undergoing a warm seclusion. Most often the near-tropopause vertical and horizontal features are far more complex than that. Still, our filtering method does capture wave structures with flat vertical tilt, so in the events where axisymmetric vortices are present, they are included within the filtered signal.

Global climate model experiments where planetary Rossby and synoptic scale waves are studied usually do not have enough vertical resolution to resolve on their own the behavior of these waves near the tropopause. The one by Fletcher and Kushner (2011) for example has 24 vertical levels, which is roughly 1.5km near the tropopause. What we see from GPS-RO observations is that planetary and synoptic-scale waves develop features that have a very short vertical scale, whereas GCMs generally tend to show a too smooth tropopause region (Gettelman et al., 2010; Hegglin et al., 2010). For this reason we are careful not to link our filtered wave signals with model studies about extratropical waves. As stated in subsection 2.3, the purpose of this paper is not to disclose particular properties of the waves, but their overall impact on the TIL (l. 250-254).

**Point-by-point responses to Ref#2**

Major Comments:
Line 278: Regarding Fig. 3: Are the tropopause zonal structures calculated directly
from the GPS-RO or from the tropopause-base coordinate? Does it make any difference?
Also, if these are zonally averaged tropopause heights, it should state so in the
Fig. 3 caption.

The tropopause zonal structures in Fig. 3 are obtained from GPS-RO, which is now stated in the figure caption. Note that in the subsection 2.2, we explain that the tropopause heights are gridded as well, which serves as the base coordinate for averaging later on. No zonal structures in tropopause height are obtained from any zonal average. The zonal means in Figures 4-9, A1 and A2 are averaged with respect to the GPS-RO observed (gridded) tropopause heights.

The paper mentions summer/winter differences in discussing Figs. 5, 7, and 9, yet it
appears as if some the largest TIL events occur more in the spring, assuming the year
tick marks denote the start of the new year. Would it be useful to discuss the results in
terms of four seasons instead of just two?

The stronger TIL in late winter/spring is related to SSWs, but the overall TIL enhancement from extratropical wave modulation is well explained with the winter and summer mean tropopause based anomalies.

[Figure]

[Figure]

[Figure]

[Figure]

We calculated the autumn and spring equivalents of Figs. 4, 6, 8, A1 (e,f). Autumn tends to yield similar results as summer, while spring tends to be close to winter results, except from 80ºN where they are intermediate. We mention this now in the corresponding subsections (see lines 423, 482, 582 and 748).

Minor Comments:
Line 2: Probably should be something like "...how much amplitude and variability of the Tropopause Inversion Layer (TIL) comes from modulation by..." (optional)

Corrected. We prefer to use the expression 'strength' instead of 'amplitude' when referring to the TIL.

Line 10: Once again: "The instantaneous modulation..." of the tropopause?, the TIL? I think it would read better if it explicitly stats what is being modulated. (optional)

We specify what's modulated by the waves (the temperature field and its gradients around the tropopause) in the first paragraph now, so it applies to the rest of the abstract when we refer to modulation again.

Line 12: "...minor importance for the TIL amplitude and variability..." (optional)

Corrected.

Line 29: "...all latitudes..." The paragraph begins with discussing the extratropical TIL so does "all latitudes" include the tropics too or just extratropics?

Indeed it also includes the tropics. We substituted 'all latitudes' for 'globally'.

Line 120: The paragraph beginning on line 120 starts with ERA-Interim and ends discussing GPS-RO. It seems awkward. Also, this appears to be the only mention of the ERA-Interim winds. It might improve the reading to tell how the ERA-Interim winds will be used to place the GPS-RO observations into a broader context by identifying SSW events and seasonal wind changes. It also might be a good idea to acknowledge the source of the winds explicitly in the appropriate figure captions as well. (optional)

The last paragraph of subsection 2.1 was split into two: the second refers to ERA-Interim exclusively and explains the purpose of the use of reanalysis winds: to show seasonality and polar vortex disturbances, complementing GPS-RO information.

We added ERA-Interim as the source for the winds in the caption of Fig. 5 (this wasn't necessary in the captions of Figs. 7, 9 and A2, since they refer back to Fig. 5 already).

Line 165: "...all..." Weather sometimes changes more rapidly than 2-days. Maybe the "all" should be qualified to "most of" or "the majority of". (optional)

Corrected (for 'most of').

Line 194: "...and frequency..." "and lower frequency" maybe

Corrected.

Corrected.

We now point to "winter and into spring" when referring to when the strongest TIL occurs in Fig. 5a. We would prefer to keep the expression 'clearly discernible': although the TIL is a narrow feature in the vertical dimension, the contrast between the lowermost stratosphere (dark blue color) and the TIL static stability values (orange to red) in winter/spring is striking even at first glance in Fig. 5a, and worth highlighting in our opinion.

We now point to "summer and into autumn" when referring to the strongest TIL occurrence in Fig. 9a. We prefer not to take out the word 'clear' for the same reason as in the previous comment.

Thank you for pointing this out, we also found the same happening with NH-SH and each acronym is properly defined now.

**Point-by-point responses to Ref#3**

Major comments
1. A key aspect of this whole paper is the recognition that the purely adiabatic impact of a wave on static stability in the tropopause region and the concomitant modification of the tropopause height may lead to a systematic TIL enhancement. Figure 2 serves as a motivation and explanation. However, I did not quite understand the related description in lines 258-260, which seems to present a key argument in this context. Please clarify.
Several times in the text (e.g. bottom of page 9) the authors stress that they typically expect a westward phase tilt with height corresponding to upward propagation of waves. This may be OK for planetary waves, but synoptic-scale waves tend to be evanescent in the stratosphere, so I would not expect much of a phase tilt in the lower stratosphere. In addition, wave modes which represent baroclinic instability (like in the Eady model) do have a westward tilt with altitude regarding perturbation geopotential or the perturbation streamfunction, but the associated tilt of the temperature anomaly actually has an eastward tilt with height (see Fig. 8.10 in the textbook of Holton 2004). Also, it has been argued in the past that some gravity wave activity is actually generated in and,

hence, emanates from the tropopause region (O'Sullivan and Dunkerton, 1995), and the phase tilt of such waves seems less clear or may be insignificant. How does this affect the arguments presented in the paper, which evoke a westward tilt with altitude?

We expanded Figure 2 (new panels 2b and 2c) and the text corresponding to lines 258-260 to better explain the wave modulation mechanism and how it affects the T profile, the local lapse-rate around the tropopause and ultimately tropopause height.

Regarding the wave vertical tilt, both our filtering method and our wave modulation mechanism apply to waves with any possible vertical tilt. We start by showing an idealized example with the commonest wave type (upward-propagating Rossby wave), but now we also explain how this can work with any wave type. We added several paragraphs in subsections 2.3 and 2.4 to clarify this.

Regarding GW and IGW generated near the tropopause: our filtering method is valid for any possible vertical phase tilt of the wave's temperature structure, but since we use a daily dataset it can only filter out waves with periods of 2 days or longer, this excludes IGW and GW that have shorter periods (see last paragraph in section 2.2).

Should GWs and IGWs have an imprint on the zonal-mean, tropopause-based N2 profile and the TIL, this would be visible in Figs. 5b, 7b, 9b, and A2b, but these figures rather point to SSWs events and the polar summer TIL.

2. Fig. 3 shows a few selected cases. Are these random picks, or are these cases where the wave signal at the tropopause level can be seen best. If the latter is true, this would mean that often the wave signal may be rather incoherent and hard to interpret?!

Fig. 3 shows cases where there is a clear modulation of the tropopause zonal structures by the wave anomalies. The picks are thus not random, but nevertheless they weren't difficult to find. During seasons where the wave signature on the TIL is strongest (mid-latitude winter and polar summer), the band corresponding to Fig. 3c shows a clear modulation of the tropopause nearly every day. Then, also almost every day, one or a couple of other bands show significant zonal ranges of tropopause modulation at the same time. We added a short caveat mentioning this in lines 367-372.

Adding to this, the mean wave signatures on the tropopause-based vertical profiles (Figs. 4-9, A1 and A2) further show evidence that the wave modulation is happening systematically.

3. In their interpretation sections, the authors could tie their results more comprehensively to earlier studies, including model studies, in particular the one by Miyazaki et al. (2010), who systematically quantified different mechanisms for TIL formation in a GCM including their hemispheric and seasonal behavior. Also, to what extent does the currently analysis (implicitly or explicitly) include the traveling cyclones and anticyclones which have been argued before to play a role for TIL formation, and to what extent are the current results consistent with the mechanisms suggested earlier? authors choose to leave this as a remaining question (see Page 12), but it could at least be formulated explicitly as a question.

We followed this suggestion. In the new section 5, we link our wave modulation mechanism to isentrope packing above anticyclones (Wirth 2003, 2004), wave breaking (Erler and Wirth 2011) and radiative effects from water vapor / clouds (Miyazaki 2010, Kunkel et al. 2016,

Ferreira et al. 2016). We suggest that our wave modulation mechanism integrates some previously proposed TIL enhancing mechanisms, rather than being a completely different dynamical feature in the tropopause region.

4. I do not like the use of the terms 'warming' and 'cooling'. The authors consider band-with filtered wave signals, in other words anomalies from the zonal mean. If a plot shows a local warm or cold anomaly, this does not necessarily imply that there is/was warming or cooling. It could just as well be the result of horizontal advection, i.e. the original air was replaced by warmer or colder air, but this does not imply any warming or cooling (neither diabatic nor adiabatic). Using more precise terminology would make the discussion of the processes more lucid.

   We removed the following sentence from the first paragraph in section 4:
*"When using the terms cooling/warming, we refer to the net effect of extratropical waves on the tropopause-based zonal mean profile, since certain levels are cooler/warmer in the seasonal mean profile due to these waves. "*
   … and we substituted the terms warming and cooling for "warm anomaly" and (mean) "cold anomaly" throughout the manuscript.

Minor issues
1. Line 86: replace "high amounts of ...." by a more idiomatic expression.
   Replaced for "large amounts".

2. Line 110: the 100 m vertical resolution of GPS-RO temperature profiles, is this really comparable to the vertical resolution of radiosondes? I thought that the latter have even significantly higher resolution. To be sure, I believe that 100 m resolution is sufficient for the purpose of the current paper.
   Radiosonde data have higher vertical resolution (several tens of meters), this is true, but GPS-RO with around 100m are not far away; and the theoretical limit of the RO technique, which could be implemented in the future, is of around 40-60m. We rephrased this sentence into "The vertical resolution and height range of the COSMIC GPS-RO temperature profiles are *close to those of* radiosonde data" to avoid any disagreement.

3. Line 126: replace "grid with 10º separation" by a better expression (grid points may have a separation, but not the grid itself).
   Replaced for "regular 10º longitude grid".

4. Line 128: replace 'exponentially-folding function' by 'Gaussian function'
   Corrected.

5. Line 153: replace "+- 1 longitude grids" by a better expression (and same with time-discretization)
   replaced for "... are filled by averaging neighboring grid points (first +-1 in longitude, then +-1 in time)".

Corrected.

Corrected.

Corrected.

Quasi-geostrophic theory is an approximation that uses the shallow water equations, so this is already implied when we refer to the wave as a disturbance propagating in a quasi-geostrophic flow.

We now refer to the $f^2/gh$ term as "*the vertical wavenumber describing the vertical propagation of the wave in terms of equivalent depth*".

Regarding the explanation of equivalent depth (the mean depth of the thin layer of fluid on a sphere in the shallow water system of the quasi-geostrophic approximation), we feel that we already describe the used dispersion relation of the Rossby wave in detail. For more, the readers are referred to the book by Andrews et al. (1987).

We feel that a discussion of whether the quasi-geostrophic approximation can be used to describe waves in the stratified atmosphere is not necessary, since it serves only as a simple motivation and is not used in our study at any point. We state several times that our manuscript is focused on the TIL and its enhancing mechanisms, not on particular properties of atmospheric waves.

The use of this idealized approximation to define the Rossby wave is motivated in the two paragraphs that follow: even with this very idealized case, the dispersion relation can be in any place within the wavenumber-frequency domain, depending on the background winds. We now specifically state that taking into account more processes (meridional propagation, horizontal/vertical wind shear), the picture gets even more difficult → thereby the choice of broad filter regions instead of following a specific dispersion curve (which is futile since it would only capture the waves occurring within a certain range of zonal winds!).

10. Bottom of page 7: I find it misleading to talk here about "different wave types". As the authors show quite clearly, there is a broad spectrum of waves, which cannot be classified into different types as in the tropics. This is, by the way, exactly what the authors say themselves on page 10 ("unable to differentiate particular wave types"). Maybe one should simply refer to "different parts of the spectrum".

We followed this suggestion: when we refer to our filtered signals (the different wavenumber-frequency domains defined in Fig. 1, and the resulting filtered anomalies) we replaced "wave type(s)" for "wave spectrum region(s)" throughout the paper.

The references to equatorial waves or specific extratropical waves (e.g. baroclinic Rossby wave) that are not part of our results are kept as "wave types" since they classify as such.

11. Line 236: replace "is outstanding" by a more idiomatic expression.
Replaced for "...their filtered signal stands out from the (unavoidable to filter) background noise".

12. Line 249: replace "associated to" by "associated with".
Corrected.

13. Line 267: replace "would remain the same despite the presence of wave anomalies" by "are unaffected by the wave anomalies"
Corrected.

14. The longitude-height sections of Fig. 3 could be improved by actually ploting all the way from -180° to +180° .
Our longitude grid goes from -175º to 175º (with 10º spacing between grid points): there is no zonal grid point missing in Fig. 3 and all the information is within the plots. Unfortunately the plotting function adjusts to the zonal grid and does not interpolate between both ends to close the circle. Note that we cannot use a grid starting at -180º and ending at 180º for the space-time filtering: both ends are on the same meridian and would lead to errors in the filtered signals because one grid point would be repeated.

15. Line 360: In what sense it s = 0 a wave?
It represents zonally-symmetric, annular oscillations in time (any possible pole-trapped wave mode). We include this now in lines 245-246.

16. Line 511: I would try to avoid the abbreviation "mSSW".
We substituted this abbreviation for "major SSW" throughout the manuscript.

17. Line 611: replace "humbler" by "smaller"
Corrected.

18. Figure 2: put axes labels, i.e. longitude (or λ) and altitude (or z).
This is surprising to us since Figure 2 in our submitted manuscript (and several previous draft versions that we checked) had both labels, which are somehow missing in the version with the ACPD header. We will contact the Copernicus support team about this issue if still present in the new version.

19. In figure 5 (and all corresponding figures showing time versus altitude) one might consider plotting only up to 30 km, without changing the vertical extent of the plot. This would give somewhat better resolution in the vertical, putting more emphasis on the features around the tropopause region, which is relevant here. On the other hand, the information way up in the stratosphere is not very relevant for this paper (except where discussing stratospheric warmings).

Figs. 7 and 9 show major/minor/final stratospheric warmings, so the proposed change in the vertical scale would affect only Figs. 5 and A2. The visibility of the TIL in all these figures is good already, and we advocate to keep the same format for all Figures 5, 7, 9, and A2 for consistency throughout the paper.

**Point-by-point responses to interactive comment by A. P. Ferreira**

Specific comments
Line 4: the phrase "it also puts other TIL enhancing mechanisms into context" should read as "it also puts other TIL enhancing dynamical mechanisms into context", since radiative effects are not addressed in the study.

It is true that the wave modulation (dynamical) mechanism is investigated in our paper, but when we subtract the wave anomalies (Figs. 5b, 7b, 9b, and A2b) the TIL that remains there can be due to any other mechanism (dynamical or not), including radiative or any diabatic processes, whose overall contribution to the TIL is by extension put into context and discussed throughout section 4 (and also 5 now). Therefore we prefer to keep this sentence as is.

Lines 10-11: instead of "The instantaneous modulation by planetary and synoptic-scale waves is almost entirely responsible for the TIL in mid-latitudes", it would be more accurate to say: "Planetary and synoptic-scale waves are almost entirely responsible for the instantaneous modulation of the TIL in mid-latitudes". In this way, the role of radiation is not left out, while keeping the authors' main finding (wave modulation of the mid-latitude TIL).

We feel that the original sentence in l.10-11 already implies that wave modulation is not totally responsible for the mid-latitude TIL, and prefer to keep it as is. However, we find that your comments about the role of radiation for the mid-latitude and polar TIL are worth addressing in our paper, which we discuss in detail in section 5 now.

Lines 18-20: "After many modelling studies (...) in the last decade, our study finally identifies which processes dominate the extratropical TIL strength and their relative contribution, by analyzing observations only." The sense of this sentence is questionable because the present and previous works on the extratropical TIL have dealt with different time and/or spatial scales, so they are not easily comparable; besides the FDH sensitivity tests supporting the radiative hypothesis are a mix of observations and radiative modelling. I am sure that the authors do not want to say that the previous theories are all marginal to the subject. Then, it would be more constructive to write something like, "In addition to the TIL enhancing mechanisms proposed by modeling studies in the last decade, our study now identifies which dynamical processes dominate the zonal-mean extratropical TIL strength and their relative contribution, by analyzing observations only."

This is a fair point, perhaps this paragraph came out too strong. We rephrased the beginning of the paragraph containing l.18-20 to make it more inclusive.

Our point of view is that our wave modulation mechanism rather integrates several previously proposed mechanisms: particularly radiative effects from water vapor and clouds, and the isentrope packing above anticyclones (e.g. Wirth 2003, 2004), they should be included in the filtered wave signal since they travel with the wave and its embedded cyclones-anticyclones. We discuss this in detail within the new section 5.

Lines 363-364: it would be helpful to the reader to clarify the expression "TIL enhancement" (from the values given in the text, it refers to the increase of $N^2_{max}$). Since it is used more than once, it should be explained in the introductory part of section 4.

We added a specification in the first paragraph of section 4.1 linking the N2 increase above the tropopause to the TIL enhancement.

Lines 380-381: when saying that the TIL is "almost completely gone" after subtracting the extratropical wave signal, the authors meant to say that $N^2_{max}$ is greatly reduced. This might be clarified in parenthesis.

We feel that an additional clarification after this sentence is not necessary: the reduction in N2 is specified in detail right after this sentence in the remainder of the paragraph.

Lines 501-502: "The only other mechanism restricted to polar summer that could enhance the TIL is water vapor radiative cooling of the tropopause". The word "restrictive" seems misleading here. I would suggest rephrasing in this way: "The only other mechanism that could enhance the polar summer TIL is water vapor radiative cooling of the tropopause".

We rephrased this sentence into: "The only other known mechanism that could enhance the TIL in polar summer is water vapor radiative cooling of the tropopause". We also made the same change in the Concluding Remarks (sec. 6) in a phrase which used a similar expression.

Lines 559-560: Since the text refers specifically to "the remaining TIL", I would suggest to replace "is enhanced by" –> "is due to"

We followed this suggestion.

Figures 5, 7, 9: in Figure 5, we expected to see $N^2$ values in the range 1-2 x10^-4 s^-2 in the upper troposphere. So, I don't understand why $N^2$ is shown in white (or is not shown) below the tropopause. The same applies to Figures 7 and 9, even if the high-latitude upper-troposphere static-stability values are different. I guess this was done to enhance the color map in the lower stratosphere. Anyhow, values not shown in the plots should be mentioned at some point (e.g., figure caption of Figure 5).

We added a specification in Fig. 5 caption that tropospheric N2 values are left blank. Figs. 7, 9, and A2 later refer back to Fig. 5.

[revised manuscript text omitted]

---

## Referee Report (RR1)

ACPD REVIEW

TITLE: Wave Modulation of the Extratropical Tropopause Inversion Layer
AUTHORS: Robin Pilch Kedzierski, Katja Matthes, and Karl Bumke

Summary:  This was a strong manuscript during the initial review and
has been improved further in revision. The writing is very clear. The
context of the effects of the TIL (Tropopause Inversion Layer) on wave
propagation and transport issues is presented in the introduction. The
GPS RO data description, averaging, and filtering is comprehensive and
detailed. This paper examines the influence of waves on the middle
latitude and polar tropopause TIL structure. Results show that most of
the extratropical TIL variability can be explained by waves with 4–25
day periods. The need for further modeling and observational research
to more precisely identify the wave structures, amplitudes, and origin
are identified where appropriate in the text. Close to the pole, 80N,
the waves are smaller scale and more difficult to identify.
Nevertheless the seasonal changes at 80N and the difference season
cycle between the polar regions and middle latitudes can be seen.
Also noteworthy are figures illustrating downward propagation of
stratospheric warming events as an additional factor influencing the
polar TIL. The influence of radiation and radiative–dynamical feedback
on the TIL is discussed as a likely mechanisms to explain the residual
signal not captured by the GPS wave analysis. Overall this study makes
excellent use of the high vertical resolution GPS observations to
characterized the TIL.

Overall Recommendation:
Publish.

---

## Referee Report (RR2)

**Review of the revised manuscript** "Wave Modulation of the Extratropical Tropopause Inversion Layer" by R. P. Kedzierski, K. Matthes and K. Bumke, submitted for publication to ACP

**Recommendation:** Some further revisions

The authors have carefully revised the manuscript, and many of my issues have been clarified in a satisfactory manner.

Yet, clarification had also the effect that some new issues surfaced, which were not evident to me from the original version. Most of my new issues refer to interpretation and terminology, so it should be possible to address them by appropriate rewording of specific sections. Just to be sure: the work done is valuable and the paper should be published eventually.

**Major issues**

1. I welcome the new discussion (section 5 in the revised manuscript), because amongst others it puts the work into the context of previous work. The authors say that their proposed "mechanism" is not a "completely separate dynamical feature" [a somewhat awkward language], but that rather "earlier proposed mechanisms are integrated in their mechanism" (lines 660 ff). In my view this somehow conflicts with the claim that they "introduced a new mechanism" (line 626, also line 20 "our wave modulation mechanism").

   Maybe I start here to quibble about terminology, but what is a "mechanism" versus or a "feature" in the authors' opinion?

   To me, what the authors show boils down to conservative reversible dynamics on (typically) synoptic time-scales and its effect on the zonally averaged TIL in a tropopause-based coordinate system. This is what the method sees, because the method considers spatio-temporal deviations from an average (i.e. the wavenumber-frequency filtering). The authors call this "wave modulation". I find this somewhat misleading, because obviously the method applied cannot see anything but waves; even a very localized anomaly (a delta function) shows up as a superposition of waves (maybe this is alluded to on lines 291–295, but I did not understand this paragraph). In other words, a very local displacement of the tropopause (say, through an isolated anticyclone) does produce the effect studied in the paper, but in my view "wave modulation" is not a good term to describe the associated mechanism. If I played the devil's adocate I would say that the authors "merely" study an old mechanism with a new diagnostic approach and new data (which, after all, I consider to be very valuable!).

   What *is* important in this paper is the recognition that local (in space and time) reversible modulation of the tropopause region ("adiabatic transient dynamical effect.. ", line 635) creates a TIL in the tropopause-based average, in a certain sense by purely "kinematic" reasons. To be sure, this is implicitly contained in some of the earlier work, but the current analysis brings out the point quite clearly and provides an observational foundation. I think this is the most original part of the paper, and this should be pointed out more clearly.

2. An important part of the argument is contained in figure 5, where panel (a) shows the "full" signal and panel (b) shows the same except with "the daily extratropical wave signal subtracted". What is the "daily extratropical wave signal" and how does this relate to the different spectral subspaces defined earlier? This must be very clearly stated, because it is the basis of the authors' claim that in midlatitudes the TIL is mostly due to "the wave modulation mechanism".

   Related to this issue: what is "instantaneous modulation" in line 484? After all, the maximum temporal resolution in this analysis is one cycle per day, so the analysis is blind to anything operating on a faster scale. It, therefore, seems difficult to make statements about "instantaneous" effects.

3. Is the conclusion on lines 679–681 correct, namely that "the zonal mean seasonal TIL is dominated by waves of synoptic and planetary scales"? After all, the current method is blind regarding fast (sub-daily) gravity waves. How can one then say with confidence that the zonal-mean seasonal TIL is dominated by synoptic and planetary scales? Wouldn't such a statement require a method that includes both types of waves and then allows one to quantitatively estimate their relative importance?

**Minor issues**

1. I am still not happy with the uncommented use of the shallow water dispersion relation. In their reply the authors state that "quasi-geostrophic theory is an approximation that uses the shallow water equations". I do not agree with this statement. What the authors probably mean is that for linear wave solutions one usually separates the vertical dimension from the horizontal plus time dimension, which after separation of variables effectively leads to a shallow water equation for each vertical mode.

2. The argument in line 218 seems nonlogical: the authors say that if there is no meridional propagation (as they assume), one can set $l = 0$. However, this is not strictly true: there may well be situations with no meridional propagation but still $l \neq 0$, e.g. when $\psi' \propto \cos(ly)$. Of course the reverse is true: if $l = 0$, there is no meridional propagation.

3. Line 273, "extratropical waves have vertical tilts in their temperature structures": I do not think this is generally true. How about synoptic scale waves which are trapped in the troposphere? Shouldn't they be characterized by zero vertical tilt at the tropopause level?

4. Line 307, "commonest": shouldn't this read "most common"?

5. Line 295, what does "filtered out" mean? Does it mean that this is left out (i.e. omitted) by the authors methodology and, therefore, not present in the remaining analysis? Or does it mean that it is accounted for by the filter (i.e. included in the filter) used and, thus, present in the remaining analysis?

6. I do not find the new term "spectrum region" very well chosen. How about "spectral region" or "spectral subspace" or "part of the spectrum"?!

---

## Author Response (AR2)

**Response to reviews of Pilch Kedzierski et al.: "Wave Modulation of the Extratropical Tropopause Inversion Layer"**

Dear Editor,

We would like to thank the reviewers for the helpful and encouraging comments. In the following paragraphs we include our point-by-point response to each comment in the reviews along with the changes made in the manuscript. The referee's comments are in blue font, and our replies are in normal font. Every change made in the revised manuscript is highlighted. We hope that the new version fulfills the reviewer's requests.

Yours sincerely,

Robin Pilch Kedzierski
Katja Matthes
Karl Bumke

**Point-by-point responses to Ref#3 (report #2)**

Major issues
1. I welcome the new discussion (section 5 in the revised manuscript), because amongst others it puts the work into the context of previous work. The authors say that their proposed "mechanism" is not a "completely separate dynamical feature" [a somewhat awkward language], but that rather "earlier proposed mechanisms are integrated in their mechanism" (lines 660 ). In my view this somehow conflicts with the claim that they "introduced a new mechanism" (line 626, also line 20 "our wave modulation mechanism").
Maybe I start here to quibble about terminology, but what is a "mechanism" versus or a "feature" in the authors' opinion?
To me, what the authors show boils down to conservative reversible dynamics on (typically) synoptic time-scales and its effect on the zonally averaged TIL in a tropopause-based coordinate system. This is what the method sees, because the method considers spatio-temporal deviations from an average (i.e. the wavenumber-frequency filtering). The authors call this "wave modulation". I find this somewhat misleading, because obviously the method applied cannot see anything but waves; even a very localized anomaly (a delta function) shows up as a superposition of waves (maybe this is alluded to on lines 291-295, but I did not understand this paragraph). In other words, a very local displacement of the tropopause (say, through an isolated anticyclone) does produce the effect studied in the paper, but in my view "wave modulation" is not a good term to describe the associated mechanism. If I played the devil's advocate I would say that the authors "merely" study an old mechanism with a new diagnostic approach and new data (which, after all, I consider to be very valuable!).

What is important in this paper is the recognition that local (in space and time) reversible modulation of the tropopause region ("adiabatic transient dynamical effect.. ", line 635) creates a TIL in the tropopause-based average, in a certain sense by purely "kinematic" reasons. To be sure, this is implicitly contained in some of the earlier work, but the current analysis brings out the point quite clearly and provides an observational foundation. I think this is the most original part of the paper, and this should be pointed out more clearly.

-We substituted the term 'dynamical feature' for 'dynamical process' for better accuracy. We clarify now that the integration of mechanisms proposed in earlier literature into our wave modulation mechanim is by interpreting their transient effects in lines 664-665.

-We feel that the naming of our TIL-enhancing mechanism, 'wave modulation', is accurate since it's done by (planetary+synoptic scale) waves which modulate the UTLS temperature and N2 zonal and vertical structures. In fact, in our earlier paper 'The tropical TIL: variability and modulation by equatorial waves' we originally referred to this mechanism as 'wave forcing' but were asked to rename it into 'wave modulation', which is more in line with how the mechanism works.

-We rephrased lines 297-298 to better explain that if local anomalies in the gridded GPS-RO data are not part of a wave (like the mentioned isolated anticyclone), then they are not filtered, and thus not part of the wave anomalies that are analyzed in our study.

-We agree that the important point about the wave modulation mechanism, that it is transient and reversible, is perhaps not highlighted enough in parts of the manuscript. Whereas this is mentioned consistently throughout sections 2.3, 2.4 and 5, this was not the case in the abstract, 1-introduction and 6-concluding remarks. We rephrased/added a few sentences in the latter sections to make the reader more familiar with the difference of our mechanism with previous literature, which focused more on permanent and irreversible effects. See lines 11, 85-87, 733-735.

2. An important part of the argument is contained in figure 5, where panel (a) shows the "full" signal and panel (b) shows the same except with "the daily extratropical wave signal subtracted". What is the "daily extratropical wave signal" and how does this relate to the different spectral subspaces defined earlier? This must be very clearly stated, because it is the basis of the authors' claim that in midlatitudes the TIL is mostly due to "the wave modulation mechanism".
Related to this issue: what is "instantaneous modulation" in line 484? After all, the maximum temporal resolution in this analysis is one cycle per day, so the analysis is blind to anything operating on a faster scale. It, therefore, seems difficult to make statements about "instantaneous" effects.

-In Figs. 5b, 7b, 9b and A2b the filtered anomalies of all the defined spectral subspaces were subtracted. 'Daily' referred to the time resolution of the GPS-RO gridded data and the filtered wave anomalies. We removed the term 'daily' when referring to the subtraction of the filtered wave anomalies in the caption of Fig. 5 and throughout section 4 to avoid any conflict since it is not related to the defined spectral subspaces.

-Regarding 'instantaneous modulation', this referred to the absence of permanent effects from our filtered wave anomalies. We substituted this expression for 'transient modulation' throughout the manuscript for better clarity.

3. Is the conclusion on lines 679-681 correct, namely that "the zonal mean seasonal TIL is dominated by waves of synoptic and planetary scales"? After all, the current method is blind regarding fast (sub-daily) gravity waves. How can one then say with confidence that the zonal-mean seasonal TIL is dominated by synoptic and planetary scales? Wouldn't such a statement require a method that includes both types of waves and then allows one to quantitatively estimate their relative importance?

Once the filtered wave signals are subtracted, the gridded dataset remains containing wave variability with frequencies greater than 96 days or lower than 2 days. Therefore, any contribution to the zonal-mean, tropopause-based N2 profiles coming from IGW or GW (or other processes) is visible in Figs. 5b, 7b, 9b and A2b. In these plots, the remaining TIL is weak or nearly absent, therefore we conclude that the planetary and synoptic-scale waves dominate, which is discussed throughout section 4.

Minor issues
1. I am still not happy with the uncommented use of the shallow water dispersion relation. In their reply the authors state that "quasi-geostrophic theory is an approximation that uses the shallow water equations". I do not agree with this statement. What the authors probably mean is that for linear wave solutions one usually separates the vertical dimension from the horizontal plus time dimension, which after separation of variables effectively leads to a shallow water equation for each vertical mode.

We now call it 'shallow-water dispersion relation' instead of just dispersion relation.

2. The argument in line 218 seems nonlogical: the authors say that if there is no meridional propagation (as they assume), one can set $l = 0$. However, this is not strictly true: there may well be situations with no meridional propagation but still $l \neq 0$, e.g. when $\psi' \propto \cos(ly)$. Of course the reverse is true: if $l = 0$, there is no meridional propagation.

We rephrased the sentences regarding meridional propagation, simply stating that it is neglected. In the $\psi' \propto \cos(ly)$ case, the term representing meridional flow would counteract the meridional wavenumber, amounting to zero in the dispersion relation anyways. In lines 235-242 we already highlight that the dispersion relation is used as an idealized and illustrative case to show the effect of background zonal winds on the Rossby wave placing within the w-k domain, which complicates further when meridional propagation and horizontal-vertical wind shear (more realistic conditions) are taken into account.

3. Line 273, "extratropical waves have vertical tilts in their temperature structures": I do not think this is generally true. How about synoptic scale waves which are trapped in the troposphere? Shouldn't they be characterized by zero vertical tilt at the tropopause level?

Upward-propagating planetary Rossby waves, inertia-gravity waves or tropospheric synoptic-scale baroclinic waves are the most common wave types found in the extratropics and their structures do have vertical tilts. However, in section 2.4 we explain that our filtering method also captures the cases where there is no vertical tilt of the wave, and the wave modulation mechanism also applies to these cases (lines 296-314).

4. Line 307, "commonest": shouldn't this read "most common"?

We followed this suggestion

5. Line 295, what does "filtered out" mean? Does it mean that this is left out (i.e. omitted) by the authors methodology and, therefore, not present in the remaining analysis? Or does it mean that it is accounted for by the filter (i.e. included in the filter) used and, thus, present in the remaining analysis?

The intended meaning was that the waves were filtered out of the gridded data. We removed the word 'out' to avoid confusion (l. 297 and 638).

6. I do not find the new term "spectrum region" very well chosen. How about "spectral region" or "spectral subspace" or "part of the spectrum"?!

We substituted 'spectrum regions' for 'spectral regions' throughout the manuscript.

[revised manuscript text omitted]

---

## Author Response (AR3)

**Response to Co-Editor Decision: Publish subject to technical corrections. Pilch Kedzierski et al.: "Wave Modulation of the Extratropical Tropopause Inversion Layer"**

Dear Peter Haynes,

Many thanks for accepting our manuscript for publication. We appreciate the technical comments and the chance to perform some final minor corrections. In the following paragraphs we include the changes made in the final manuscript. The editor's comments are in blue font, and our replies are in normal font. Every change made in the manuscript is highlighted.

Kind regards,

Robin Pilch Kedzierski

Katja Matthes

Karl Bumke

You may also wish to make a final check of the paper to ensure that no minor typographical or similar errors have been missed.

After performing a more thorough check on the manuscript, we found and corrected the following errors: Modelling → Modeling (lines 59 and 89); and Occuring → Occurring (line 417).

l275: 'Extratropical waves have vertical tilts in their temperature structures, and if the anomalies are large, they can effectively modulate tropopause height as explained next.' — As noted by the referee, extratropical waves do not HAVE to have vertical tilts in their temperature structures (even if many waves do), and what you have written here might be interpreted (by a casual reader) as saying that waves affect tropopause height ONLY if they have vertical tilts in their temperature structures, which I don't believe is true. So personally I would simply omit the reference to 'vertical tilts'.

The sentence now reads: 'If the extratropical wave's temperature anomalies are large, they can effectively modulate...' (line 276).

l710: 'By tropopause-based zonal averaging of these
signals at certain latitude bands, we were able to quantify how much of the extratropical TIL at mid and polar latitudes is explained by the transient modulation of the tropopause region by the planetary and synoptic-scale waves.' — my understanding is that you have constructed e.g. Fig 5b by removing what you define as the wave signal from all temperature/static stability profiles to generate a new set of 'waveless' profiles, then you apply tropopause-based averaging to those 'waveless' profiles. Your statement 'tropopause-based zonal averaging of these signals' does not seem to me to be a clear description of that, since you have just used 'wave signal' meaning what is removed, not what is retained. You may wish to consider changing this statement.

This sentence in particular referred to Figs. 4, 6, 8 and A1; which is now specified (line 713). In the next sentence we refer to the 'waveless' profiles where subtraction has been performed, and we specify now that this applies to Figs. 5, 7, 9 and A2 (line 714).

[revised manuscript text omitted]